



# Biosphere-atmosphere related processes influence trace-gas and aerosol satellite-model biases.

Emma Sands[1], Ruth M. Doherty[1], Fiona M. O'Connor[2,3], Richard J. Pope[4,5], James Weber[6], and Daniel P. Grosvenor[2,4]

[1]School of Geosciences, University of Edinburgh, Edinburgh, United Kingdom
[2]Met Office Hadley Centre, Exeter, United Kingdom
[3]Department of Mathematics & Statistics, Global Systems Institute, University of Exeter, United Kingdom
[4]School of Earth and Environment, University of Leeds, Leeds, United Kingdom
[5]National Centre for Earth Observation, University of Leeds, Leeds, United Kingdom
[6]Department of Meteorology, University of Reading, United Kingdom

**Correspondence:** Emma Sands (e.g.sands@ed.ac.uk)

**Abstract.** Biogenic volatile organic compounds (BVOCs), such as isoprene, impact aerosols, ozone and methane, adding uncertainty to assessments of the climate impacts of land cover change. Recent UK Earth System model (UKESM) developments allow us to study how various processes impact biosphere-atmosphere interactions and their implications for atmospheric chemistry, while advances in remote sensing provide new opportunities for assessing biases in isoprene alongside formaldehyde and aerosol optical depth (AOD).

The standard setup of UKESM1.1 underestimates the regional formaldehyde column by up to 80%, despite positive isoprene biases of over 500%. Seasonal average AOD values are underestimated by over 60% in parts of the Northern Hemisphere but overestimated (>180%) in the Congo.

The effects of several processes are studied to understand their impacts on satellite-model biases. Of these, changing from the default to a more detailed chemistry mechanism has the greatest impact on the simulated trace gases. Here, the isoprene lifetime decreases by 50%, the formaldehyde column increases by >20%, whilst reductions in upper-tropospheric oxidant mixing ratios decrease sulphate nucleation ($-32\%$). Organically-mediated boundary layer nucleation and contributions to aerosol mass from isoprene oxidation decrease AOD values in the Northern Hemisphere, while revised BVOC emission factors and land cover representation affect the emissions of BVOCs and dust.

The combination of processes substantially affects regional model-satellite biases, typically decreasing isoprene and AOD and increasing formaldehyde. We find significant differences in the aerosol direct radiative effects (+0.17 W m$^{-2}$), highlighting that these processes may have substantial ramifications for impact assessments of land use change.





## 1 Introduction

Biogenic volatile organic compounds (BVOCs) are emitted by plants during photosynthesis and in response to stresses such as insect infestations or droughts (Tani and Mochizuki, 2021). When BVOCs oxidise in the atmosphere, they can influence ozone ($O_3$) concentrations and methane ($CH_4$) lifetime and concentrations (e.g., Pacifico et al., 2009), as well as secondary organic aerosol (SOA) mass and number (e.g., Scott et al., 2014). As both $O_3$ and $CH_4$ are greenhouse gases and aerosols affect the radiative balance through aerosol-radiation interactions and aerosol-cloud effects, changes in BVOC emissions impact the

climate (Artaxo et al., 2022; Boy et al., 2022; D'Andrea et al., 2015; Heald and Geddes, 2016; Scott et al., 2017; Weber et al., 2022). However, land-cover-change radiative effects in general, and the radiative impacts of changes in aerosols in particular, are associated with substantial uncertainty.

Heald and Spracklen (2015) suggested a global mean pre-industrial to present-day radiative forcing of $-0.21$ W m$^{-2}$ (range of $-0.28$ to $-0.14$ W m$^{-2}$) from atmospheric composition changes driven by land use change. Decreases in biogenic SOA

contributed $+0.034$ W m$^{-2}$ ($+0.012$ to $+0.056$ W m$^{-2}$). Another study found land use driven changes in BVOC emissions between 1850 and 2000 led to a similar positive aerosol direct effect of $+0.02$ W m$^{-2}$ combined with $+0.008$ W m$^{-2}$ from aerosol-cloud interactions (Scott et al., 2017). Furthermore, an analysis of millennial land cover change impacts on SOA calculated an aerosol direct effect of $+0.022$ to $+0.163$ W m$^{-2}$ and an aerosol indirect effect between $-0.008$ and $-0.056$ W m$^{-2}$ for the period 1000 to 2000 (D'Andrea et al., 2015). This range of previously calculated land cover change radiative

forcing estimates is partially driven by the differences and uncertainties in process representation in the models used. We use a detailed model-observation comparison to identify key processes and facilitate experimental design for future studies of land cover impacts on atmospheric composition.

Recent developments in remote sensing have enabled the retrieval of isoprene ($C_5H_8$), the globally dominant BVOC (Sindelarova et al., 2014), from space (Fu et al., 2019; Palmer et al., 2022), greatly increasing the geographical coverage of BVOC

observational data compared to local measurements. This has allowed regional and global studies using $C_5H_8$ measurements (Fu et al., 2019; Palmer et al., 2022; Sands et al., 2024; Weber et al., 2021; Wells et al., 2020, 2022), instead of the BVOC oxidation product formaldehyde (HCHO). While HCHO has often been used to estimate BVOC emissions and concentrations (Kefauver et al., 2014; Marais et al., 2012; Stavrakou et al., 2015; Millet et al., 2008; Palmer et al., 2006; Strada et al., 2023), uncertainties associated with the presence of other, notably anthropogenic, HCHO sources, e.g., from combustion, and

uncertainty in the representation of the chemistry which converts $C_5H_8$ to HCHO, introduce challenges to the interpretation of the HCHO satellite data. The $C_5H_8$ satellite observations offer new opportunities for large-scale studies of $C_5H_8$ spatial distributions, seasonality and trends, which can also be used to evaluate model representations of BVOCs. While combining $C_5H_8$ and HCHO data can provide insights into BVOC oxidation and gas phase products, satellite aerosol optical depth (AOD) measurements may respond to changes in SOA. AOD is an indicator of the amount of aerosol in the atmospheric column,

which takes advantage of the attenuation of light by aerosols through either absorbance or reflectance (Wei et al., 2020).

BVOCs, HCHO and AOD are simulated in the United Kingdom Earth System Model (UKESM1.1, Mulcahy et al. (2023)). The continuous development of the model since the release of v1.1 has resulted in the improved ability to test various processes





influencing the impact of biosphere-atmosphere interactions on atmospheric chemistry and aerosols, as well as sensitivity to land cover, hereafter referred to as biosphere-atmosphere processes. Some of these processes have not yet been evaluated in
the context of BVOC impacts on atmospheric composition as outlined below.

Recently, the model chemistry reaction rate constants were updated with the latest published IUPAC data. Additionally, an alternative chemistry mechanism based on the Common Representative Intermediates (CRI) v2.2 mechanism (Jenkin et al., 2008, 2019) with more detailed simulation of BVOC oxidation and $HO_x$(=$OH+HO_2$) recycling (CRI-Strat 2, (Archer-Nicholls et al., 2021; Weber et al., 2021)) has been developed within UKESM. The addition of $HO_x$ recycling increases OH concentra-
tions in low-$NO_x$ environments characterised by high BVOC emissions, which has significant implications for the oxidative capacity of the atmosphere (e.g., Archibald, 2011; Khan et al., 2021; Lelieveld et al., 2008).

Changes in atmospheric oxidation have previously been found to impact secondary aerosol formation, and, therefore, the aerosol size distribution, in some Earth system models (Karset et al., 2018; O'Connor et al., 2021; O'Connor et al., 2022). Consequently, any change to the gas phase oxidation of aerosol precursors, such as updated reaction rates or a more detailed
chemistry mechanism, may alter the impacts of biosphere-atmosphere processes for both trace gases and aerosols. Use of the CS2 mechanism, as opposed to the standard chemical mechanism StratTrop v1.0, has been studied for a range of trace gases (Archer-Nicholls et al., 2021; Weber et al., 2021), but the impacts on aerosols have yet to be assessed against observations. The sensitivity of biosphere-atmosphere processes to the reaction rate updates has also not been studied.

Another recent UKESM development is the option to include the production of an inert tracer from $C_5H_8$ oxidation that con-
tributes to organic aerosol mass through condensation pathways (Weber et al., 2022). Aerosol simulated over the tropical forest in UKESM1.0 was found to be in poor agreement with observations compared to other models (Blichner et al., 2024), possibly due to the lack of production of SOA from $C_5H_8$ in its standard setup. Further enhancements to the representation of aerosols include the addition of an organically-mediated boundary layer nucleation (BLN) scheme and updated aerosol hygroscopicity values to better match published literature (Fanourgakis et al., 2019; Petters and Kreidenweis, 2007; Ranjithkumar et al., 2021;
Schmale et al., 2018). The outdated hygroscopicity values were a potential driver of the anomalously low effective radiative forcing from aerosol-cloud interactions from organic aerosol in UKESM1.0 (Thornhill et al., 2021). All of these processes will affect the impacts of BVOC emissions on aerosols and may subsequently impact the aerosol radiative effects.

BVOC emissions themselves depend on the representation of land cover and the emission factors (EFs) associated with each plant functional type (PFT). Sellar et al. (2019) found that the dynamic land cover simulated by the land component of UKESM,
JULES, led to regional inconsistencies with observational land cover datasets for UKESM1.0, e.g. an overly wide distribution of forests in South America, attributed to a lack of fire disturbance. However, the implications of the resulting biases in BVOC emissions for atmospheric composition have not been studied in detail. Other work has also identified inconsistencies in the BVOC EFs after the expansion of the number of PFTs (Weber et al., 2023). Combining observational land cover with new EFs could have significant impacts on the simulated BVOC emissions.

In this study, we present a satellite observation-model comparison for $C_5H_8$, HCHO and AOD at a high temporal resolution and with consideration of vertical satellite sensitivity, which allows for a more accurate evaluation of present day simulated total column $C_5H_8$ (TC $C_5H_8$), tropospheric column HCHO (TrC HCHO) and AOD, than previously published work. We





quantify how recent developments beyond UKESM1.1 related to the representation of biosphere-atmosphere processes affect simulated column $C_5H_8$ and HCHO, as well as AOD, alongside the subsequent impacts on $C_5H_8$ lifetime, atmospheric oxi-
dising capacity, and aerosol size distribution. We further include an assessment of how these changes to process representation affect satellite-model biases and identify key implications for radiative effects. Section 2 introduces UKESM1.1 to which the new developments were applied, the experimental set up and the observational data. The results of the satellite-model comparison and impacts of the new and updated processes are discussed in section 3, before the key conclusions are summarised in section 4.

## 2    Data and methods

### 2.1    Model description

This study employs version 1.1 of the atmosphere-only configuration of the UK Earth System Model (UKESM1.1) (Mulcahy et al., 2023). The physical atmosphere component is the Global Atmosphere 7.1 science configuration of the Met Office's Unified Model (MetUM) (Walters et al., 2019), while the atmospheric chemistry is simulated in the United Kingdom Chemistry
and Aerosol (UKCA) model (Morgenstern et al., 2009; O'Connor et al., 2014) combined with the Global Model of Aerosol Processes (GLOMAP-mode) (Mann et al., 2010; Mulcahy et al., 2020). The resolution of the model is N96L85, equivalent to $1.875° \times 1.25°$ horizontal resolution (or approximately 135 km) at the Equator and 85 vertical levels from the surface up to 85 km altitude.

The simulations have fully interactive stratospheric and tropospheric chemistry, with either the StratTrop v1.0 (ST) (Archibald
et al., 2020) or CRI-Strat 2 (CS2) (Archer-Nicholls et al., 2021; Weber et al., 2021) chemistry mechanism. Key differences between the mechanisms in the context of biosphere impacts on atmospheric chemistry include: the addition, in the CS2 mechanism, of $HO_x$ recycling during $C_5H_8$ oxidation and the separation of lumped monoterpenes ($C_{10}H_{16}$) into $\alpha$-pinene and $\beta$-pinene. The baseline experiment in this work uses the ST chemistry scheme.

Emissions of $C_5H_8$ and monoterpenes are calculated using the interactive biogenic volatile organic compound (iBVOC)
scheme (Pacifico et al., 2011, 2012). The emissions of the BVOCs depend on temperature, carbon dioxide ($CO_2$), photosynthetic activity and the plant functional type (PFT) (see Table S1 for PFT specific emission factors (EFs)). In both chemistry schemes, monoterpenes form a condensable vapour (Sec_Org$_{MT}$, where MT stands for monoterpenes), which contributes to aerosol mass in all modes through irreversible condensation onto pre-existing particles. In the CS2 mechanism, the separated monoterpenes have a more explicit oxidation scheme that forms trace gases, such as peroxy radicals, carbonyl and hydroper-
oxides, in addition to Sec_Org$_{MT}$ (Weber et al., 2021, 2022). As other BVOCs are also known to contribute to particle growth, the Sec_Org$_{MT}$ yield is scaled by a factor of two to account for these missing sources of SOA (Mulcahy et al., 2020), unless otherwise stated.

The aerosol types considered are sulphate (SU), sea-salt (SS), black carbon (BC) and organic matter (OM), which are simulated in GLOMAP-mode and combined with the Coupled Large-scale Aerosol Simulator for Studies In Climate (CLASSIC)





aerosol scheme for mineral dust (DU) (Woodward et al., 2022; Mulcahy et al., 2020). The gas-phase chemistry and aerosols are coupled, so that the formation and growth of particles depends partially on the simulated trace gases that are aerosol precursors.

## 2.2 Model experiments

A series of nudged experiments, which were designed for a satellite-model comparison of the impacts of modifying various biosphere-atmosphere processes on $C_5H_8$, HCHO and AOD, was carried out. ERA-5 reanalyses temperature and wind fields 125 were used to nudge the model between vertical model levels 12 and 80 (on average between 800 m and 60 km above the surface). Unless otherwise stated, the experiments were run for 10 years (2005-2014), with the first year of model output not included in the analysis to allow for model spin up. The experiments were initialised using output from a pre-existing historical UKESM1.1 experiment.

The sea surface temperatures, sea-ice, relevant vegetation information (PFT fractions, leaf area index, canopy height) and 130 surface ocean biology (dimethyl sulphide and chlorophyll ocean concentrations) were prescribed based on fields diagnosed from an existing fully-coupled UKESM1.1 model run, unless otherwise specified. Anthropogenic emissions (Hoesly et al., 2018), biomass burning emissions (van Marle et al., 2017) and lower boundary conditions of greenhouse gases (Meinshausen et al., 2017) were prescribed as in the UKESM1.0 implementation for the Coupled Model Intercomparison Project Phase 6 (CMIP6, Sellar et al. (2020)).

Table 1 provides a summary of the model experiments. Three experiments focus on processes particularly relevant to the emission and oxidation of trace gases. Recent work has suggested that $C_5H_8$ and monoterpene EFs should be updated, to avoid inconsistencies such as the excessively large emissions from the grass PFTs (Weber et al., 2023). Exp_EF uses these new EFs (Table S1). The impact of a recent update to many reaction rates (Supplement 2) for consistency with more recent published data (IUPAC Task Group on Atmospheric Chemical Kinetic Data Evaluation, http://iupac.pole-ether.fr/, last access: 25/09/2024) is 140 tested in the short experiment Exp_RR, which produced one year of data for analysis. Finally, Exp_CS2 investigates the impact of using the more complex CS2 mechanism (sections 1 and 2.1). These 'trace gas' updates, in addition to affecting the concentrations and lifetimes of trace gases, may impact on aerosols by modifying the lifetime of aerosol precursors and aerosol formation pathways.

Another set of three simulations focuses primarily on aerosols. A recent correction to hygroscopicity values (Fanourgakis 145 et al., 2019; Petters and Kreidenweis, 2007; Schmale et al., 2018) is examined in Exp_HYG. Exp_BLN includes organically-mediated boundary layer nucleation (BLN) based on Metzger et al. (2010), which allows monoterpenes to contribute to the formation of new particles. The rate of formation depends on the concentration of sulphuric acid ($H_2SO_4$) and the vapour Sec_Org$_{MT}$ and is calculated as follows:

$$J = k \times [H_2SO_4] \times [SecOrg_{MT}], \tag{1}$$

where $k = 5 \times 10^{-13}$ cm$^{-3}$ s$^{-1}$. The final aerosol simulation, Exp_SOAi, includes the production of Sec_Org$_I$, where I stands for isoprene, from $C_5H_8$ oxidation, following Weber et al. (2022). Sec_Org$_I$ is a chemically-inert, condensable vapour equivalent to Sec_Org$_{MT}$ and is treated in the same way in the model, except for during organically-mediated BLN, where





**Table 1.** Processes and other model updates included in each sensitivity experiment. 'X' symbolises the inclusion of (an updated) process, while empty cells indicate the use of baseline values or the lack of a given process.

| Simulation | Updated BVOC emission factors | Updated reaction rates | CS2 mechanism | Organically-mediated boundary layer nucleation | Updated hygroscopicity values | Contribution to SOA from isoprene | ESA CCI land cover |
|---|---|---|---|---|---|---|---|
| Baseline | | | | | | | |
| Exp_EF | x | | | | | | |
| Exp_RR | | x | | | | | |
| Exp_CS2 | | | x | | | | |
| Exp_BLN | | | | x | | | |
| Exp_HYG | | | | | x | | |
| Exp_SOAi | | x | | | | x | |
| Exp_LC | | | | | | | x |
| Exp_LCEF | x | | | | | | x |
| Exp_STm | x | x | | x | x | x | |
| Exp_CS2m | x | x | x | x | x | x | |
| Exp_CS2mLC | x | x | x | x | x | x | x |

only $Sec\_Org_{MT}$ can participate in the nucleation process. In this experiment, the factor of two scaling of the $Sec\_Org_{MT}$ yield from monoterpenes is removed. This is because it was implemented to account for the lack of $Sec\_Org_I$ in the model and

is therefore redundant with the explicit condensation of $Sec\_Org_I$.

Another key aspect of the biosphere-atmosphere relationship is the representation of land cover, which is explored in two further simulations: Exp_LC and Exp_LCEF. In these experiments, the model-derived PFT fractions are replaced with a land cover climatology from the European Space Agency (ESA) Climate Change Initiative (CCI) land cover dataset (ESA, 2017). Only the updated land cover is included in the Exp_LC, while Exp_LCEF includes the updated land cover and the previously

discussed new EFs.

The various processes are combined in three merged simulations: Exp_STm, Exp_CS2m and Exp_CS2mLC. Exp_STm is expected to be the most similar to a future version of UKESM, Exp_CS2m isolates the impact of the chemistry mechanism choice alongside other model updates, while Exp_CS2mLC includes all studied processes and the alternative observation-based (ESA-CCI) land cover representation. All experiments are compared against the baseline simulation, which uses the standard

UKESM1.1 set up.





## 2.3 Observational data

Three satellite-remote-sensing datasets are used for comparing the model outputs with observational data. The monthly TC $C_5H_8$ data on a $0.5° \times 0.625°$ spatial resolution is retrieved using a machine-learning framework applied to measurements from the Cross-track Infrared Sounder (CrIS) on the Suomi National Polar-orbiting Partnership (Suomi NPP) satellite (Wells
et al., 2022). The satellite has a daytime overpass of around 13:30 local time (LT) and differs by 20 to 50% compared to ground-based column measurements, with a mean absolute difference of 34%. The CrIS TC $C_5H_8$ tends to be greater than the ground-based measurements for TC $C_5H_8$ values over $7 \times 10^{15}$ molec. cm$^{-2}$.

The HCHO data is the level 2 daily tropospheric column dataset provided as part of the Tropospheric Emission Monitoring Internet Service (TEMIS) (De Smedt et al., 2015; Royal Belgian Institute for Space Aeronomy). The columns are calculated
from measurements taken by the Ozone Monitoring Instrument (OMI) on the Earth Observing System (EOS) Aura satellite, and are available from October 2004 to December 2020. Aura orbits at 705 km and has an equator crossing time of 13:45 LT. The instrument can cover a 2600 km swath on the Earth's surface, with pixel sizes varying from 13 km $\times$ 24 km to 28 km $\times$ 150 km, resulting in almost daily global coverage (De Smedt et al., 2015). The data include random and systematic error terms for each retrieval discussed further in section 2.5.

Daily AOD level 2 data at 10 km $\times$ 10 km spatial resolution are available from the Moderate Resolution Imaging Spectro-radiometer (MODIS) instrument on the Aqua satellite (MYD04_L2, Levy et al. (2017)). Aqua has an equatorial crossing time of 13:30 LT. The scientific dataset chosen ('Optical_Depth_and_And_Ocean') is based on the Dark Target algorithm (Levy et al., 2013), and is available for both land and ocean surfaces. The performance of the algorithms varies regionally, but neither Dark Target nor the alternative Deep Blue product consistently outperforms the other (Sayer et al., 2014). The AOD data
are distributed on the Centre for Environmental Data Analysis (CEDA) archive on Jasmin (Lawrence et al., 2013; National Aeronautics and Space Administration, 2021).

## 2.4 Analysis

### 2.4.1 Model comparison with satellite observations

For the comparison of UKESM and satellite observations (performed for the baseline and Exp_CS2mLC simulations), 6-hourly
model outputs are used, to ensure the model and satellite data can be co-located in space and time to within three hours. When otherwise elucidating the importance of the different model processes monthly mean model outputs are used.

The TC $C_5H_8$ satellite observations are only available as a monthly product. Therefore, for each location on the model grid, the output from the closest time to 13:30 LT (the satellite overpass time) is chosen. Given the distribution of $C_5H_8$ in the model (Fig. S1) and the sensitivity of the CrIS retrieval to $C_5H_8$ in the troposphere (Fu et al., 2019), the satellite data should
be suitable for evaluating the model. The $C_5H_8$ observations are mapped onto the model grid to calculate the model-satellite biases. Consequently, a comparison of the monthly re-gridded satellite and temporally-filtered model total columns should provide a reasonable estimate of the observation-model differences.




The level 2 HCHO data allow for a satellite-model comparison that is more detailed in space and time. For each satellite retrieval, the closest model output in terms of latitude, longitude and time is identified. The satellite averaging kernels are then applied to the model data to account for the vertical sensitivity of the satellite retrieval. The average value of all satellite retrievals within a gridbox is calculated at daily temporal resolution for data that passed the screening tests: cloud radiance fraction between 0 and 0.5, solar zenith angle of the retrieval <80° and relative error values <75% of the satellite TrC HCHO.

A similar process is followed for the AOD comparison. The level 2 satellite data are regridded onto the model horizontal grid, and the closest model output in space and time is chosen for the calculation of the daily data. The 550 nm AOD is used, as it is the only wavelength available for both the MODIS and model. The chosen satellite data product is provided as a quality filtered dataset, therefore additional screening is not necessary, unlike for HCHO.

### 2.4.2 Quantifying impacts on the aerosol direct effect

To understand the impact on aerosol radiative effects of the changes in process representation studied here, we calculate the difference in the Direct Radiative Effect (DRE; due to aerosol-radiation interactions) between the simulations that include all processes with or without land cover (Exp_CS2mLC and Exp_CS2m, respectively) and the baseline simulation. We follow the recommendation of Ghan (2013) and calculate the DRE using:

$$DRE = (F - F_{clean}),  \qquad (2)$$

where $F$ is the top of atmosphere (TOA) radiative flux (downwelling − upwelling), and $F_{clean}$ is the TOA radiative flux excluding scattering and absorption by aerosols. Separate values are calculated for the shortwave, longwave and net fluxes. This approach has been previously used to decompose aerosol impacts (e.g., O'Connor et al., 2021; O'Connor et al., 2022; Weber et al., 2022).

### 2.4.3 Representation of uncertainty and error

All observational datasets have random and systematic errors. Therefore, when analysing the satellite-model differences, only results from locations where the absolute difference is greater than the mean satellite data error are deemed to be significant. However, the satellite errors vary in magnitude by the data product and the error variables are not always provided. Only the HCHO product includes an error estimate for both the random and systematic components. During the calculation of daily values (section 2.4.1), we propagate the HCHO errors, with the random error reduced depending on the number of retrievals identified for a given location. Hereafter, a conservative estimate of the error is used, as this value is not reduced further by the sample size when calculating monthly, seasonal and multi-annual mean values, despite the random error component. For $C_5H_8$, the value of 40% of the $C_5H_8$ column is used as a conservative estimate based on the mean difference between the satellite column and ground-based observations (section 2.3). For AOD, the error estimate over land is $\pm(0.05 + 15\%\tau)$, where $\tau$ is the AOD value at a given location, and is slightly less over the ocean (Sayer et al., 2014). The conservative estimate from over land is used throughout the analysis.





To identify significant differences between the model experiments, we use the Wilcoxon-Mann-Whitney test (Mann and
Whitney, 1947; Wilcoxon, 1945; Wilks, 2006). The Wilcoxon-Mann-Whitney test ranks all of the data, before comparing the
sum of the ranks for the two distributions. This non-parametric test was deemed most appropriate, as the modelled trace gas
column and AOD values are not consistently normally distributed over the globe. Hereafter, the 95% confidence interval is
used to assess whether the difference between model simulations is significant.

## 3   Results and discussion

### 3.1   Model comparison against satellite observations

Figure 1 shows the comparison between TC $C_5H_8$ in the baseline simulation and the monthly satellite-derived TC $C_5H_8$ dataset.
Although UKESM1.1 identifies the $C_5H_8$ emission hotspots in the tropics (e.g., Amazon, Congo basin and Indonesia), the TC
$C_5H_8$ biases in these regions exceed $1 \times 10^{16}$ molec. cm$^{-2}$ (>50% and typically around 150%) in December-January-February
(DJF) and March-April-May (MAM) (Fig. 1e,f). The model biases show a more mixed signal in June-July-August (JJA) and
September-October-November (SON), when the model simulates lower TC $C_5H_8$ values than observed in some regions, e.g.,
the southeastern USA (around $-0.5 \times 10^{16}$ molec. cm$^{-2}$, or $-60\%$ to $-100\%$) and the southeastern Amazon (around $-1 \times 10^{16}$
molec. cm$^{-2}$, or $-40\%$ to $-100\%$). These biases tend to have lower magnitudes than the positive differences in DJF and
MAM.

The positive biases in TC $C_5H_8$ in the tropics have been previously noted for UKESM for January, April, July and October
2013. Weber et al. (2021) found that model TC $C_5H_8$ values over parts of the Amazon were $2 \times 10^{16}$ molec. cm$^{-2}$ greater then the
observations throughout the year, but the average bias for northern South America was lower in October due to underestimated
$C_5H_8$ values in the east of the region. Negative biases in the mid- and high-latitudes of North America and Europe of $<1 \times 10^{16}$
molec. cm$^{-2}$ were also identified in that study, consistent with our results. We additionally highlight a seasonal change in the
sign of the bias in South America, particularly in the eastern Amazon. These negative biases in SON correspond to the lower
regional bias in October calculated in Weber et al. (2021).

The consistent results between both analyses suggest that the sign of the biases is relatively insensitive to the period of study
and temporal resolution of the model output. However, a comparison against model $C_5H_8$ output specifically for the satellite
overpass time (13:30 LT) reveals the biases calculated using the 6-hourly output method employed here may be underestimated
by up to $1.5 \times 10^{16}$ molec. cm$^{-2}$ (Fig. S2). The absolute differences between the 6-hourly and overpass time TC $C_5H_8$ are
largest over the tropical forests in South America and Africa (typically within 100% of the 6-hourly output), but the relative
difference is greater at the higher latitudes, where the TC $C_5H_8$ values are low compared to the tropics. For example, in the
eastern USA and western Europe the TC $C_5H_8$ value can be greater by a factor of 2 to 10. Globally, the difference between the
two model estimates based on the two sampling methodologies is on average $1.7 \times 10^{14}$ molec. cm$^{-2}$, which is around 1%
of the mean satellite-retrieved TC $C_5H_8$. It is also important to note that the negative bias in eastern South America in SON is
of a greater magnitude than the difference between the two sampling methodologies. Consequently, the 6-hourly outputs are



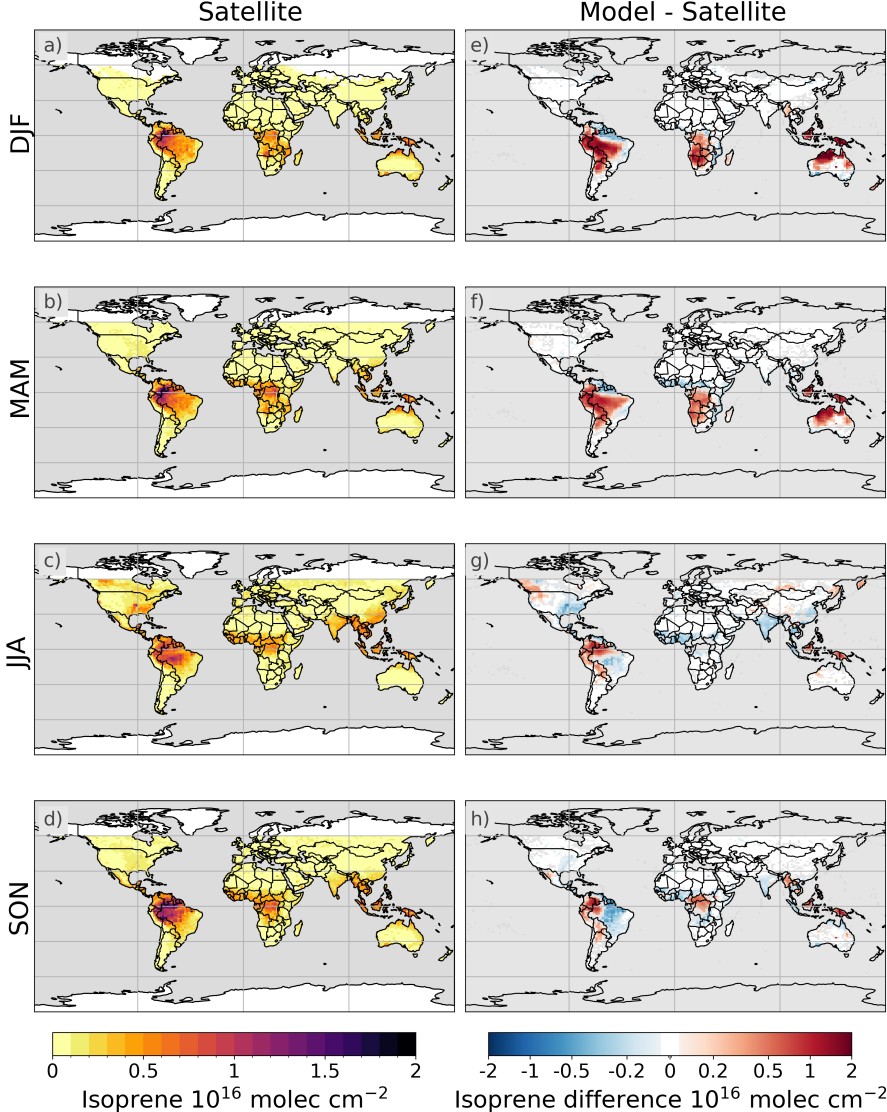

**Figure 1.** Comparison of modelled isoprene against a satellite derived isoprene dataset. The data show the mean seasonal satellite total column values for 2012-2014 (a-d) and mean seasonal differences between the baseline simulation and satellite data (model − satellite, e-h). Regions where the model-satellite difference is smaller than the 40% column value satellite error (Wells et al. 2022) have been shaded in grey. Note there are no satellite observations for January 2012 and this month is excluded from the analysis.

deemed sufficient for identifying key regions of $C_5H_8$ biases, although the magnitude of these biases may be underestimated or, in the case of regions of negative biases in the tropics, overestimated.





In contrast to the substantial positive regional biases the model displays for TC $C_5H_8$, TrC HCHO values are underestimated (Fig. 2), possibly due to limited VOC sources of HCHO in the StratTrop v1.0 chemistry scheme. Considering that recent work has shown that OMI satellite retrievals of HCHO have a negative bias compared to ground-based and aircraft observations (Müller et al., 2024), the difference between the model simulations and actual TrC HCHO may be even greater than found here. The spatial extent over which TrC HCHO is underestimated is largest in JJA, when the magnitude of the negative biases can exceed $2\times10^{16}$ molec. cm$^{-2}$ ($-50\%$ to $-90\%$) in Brazil, regions south of the Congo, the eastern USA and parts of Eurasia (Fig. 2g). The model-satellite difference in South America is of a similar magnitude in SON (continental average of $-1.05\times10^{16}$ molec. cm$^{-2}$). Over this continent, particularly in JJA and SON, the significant biases tend to occur in the east over regions of significant TC $C_5H_8$ underestimates (compare Fig. 1g,h and Fig. 2g.h), suggesting that the low model values of $C_5H_8$ may drive the negative HCHO bias in this particular region.

The model biases for AOD vary depending on season and location (Fig. 3). Negative biases occur in MAM and JJA over land in the Northern Hemisphere (NH) ($-0.1$ to $-0.4$, or $-50\%$ to $-100\%$, greater absolute magnitude in China) and over the Atlantic in Sahara dust outflow regions, where differences are greater than 0.45 (Fig. 3f,g). There is also a negative bias over the Amazon, which has the greatest magnitude in SON ($< -0.45$ in some regions, i.e., $-50$ to $-90\%$) . The strongest positive biases (relative magnitude $>90\%$) occur over the Congo (reaching over 0.45), in eastern India and Bangladesh (reaching over 0.2) and western Australia (up to 0.2). The dominance of significant negative biases is consistent with the negative annual mean AOD biases found for an ensemble of UKESM1 (Mulcahy et al., 2020) and UKESM1.1 simulations when compared to MODIS data (Mulcahy et al., 2023).

There are some differences of note between the previous studies and the methodology used for this work . Mulcahy et al. (2020, 2023) use low temporal resolution AOD output, which does not allow for sampling at the same local time as the satellite retrieval, as done here. Despite this and the slightly different periods considered (2006-2014 here and 2003-2014 in the other studies), the results are consistent. While Mulcahy et al. (2023) find a difference of $-0.032$ between the UKESM1.1 global mean AOD and MODIS observations, the equivalent value calculated here is $-0.046$. They also highlight the negative AOD bias in dust source regions (and downwind). Although the dust scheme for UKESM1.1 (Mulcahy et al., 2023) was tuned to address the negative bias found in UKESM1.0 (Checa-Garcia et al., 2021), a significant negative bias remains over the central Atlantic in MAM and JJA (Fig. 3f,g). Overall, our results suggest recent model updates to both the mineral dust and sulphate aerosol representations (Hardacre et al., 2021; Mulcahy et al., 2023) have only partially decreased the magnitude of the differences between the model and satellite observations, and the comparison results for AOD are consistent in the sign of the bias regardless of the temporal sampling frequency and period used for the data analysis.

## 3.2 Influence of changes to trace gas emission and oxidation

Three of the experiments focus on isolating the influence of BVOC emissions and oxidation in the model: Exp_EF, Exp_RR and Exp_CS2 (see section 2.2, Table 1.). The updated EFs (Table S1) drive some spatial shifts in $C_5H_8$ and monoterpene emissions (not shown). Particularly noticeable is the decrease in TC $C_5H_8$ in subtropical regions (e.g., $-70\%$ in DJF for the zonal mean at SH subtropical latitudes, Fig. 4g), where erroneously high emissions from grass PFTs have been reduced. This change



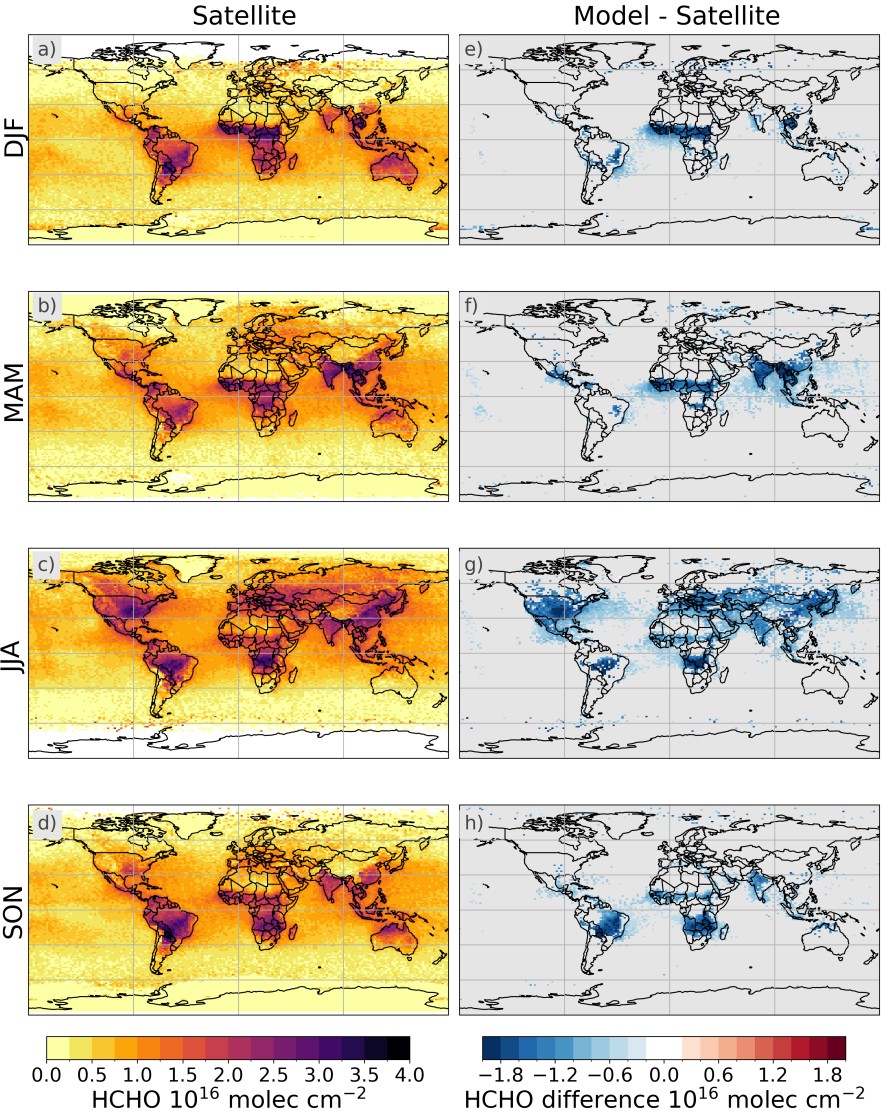

**Figure 2.** Comparison of modelled HCHO against satellite HCHO observations. The data show the mean seasonal satellite tropospheric column values for 2006-2014 (a-d) and mean seasonal differences between the baseline simulation and satellite data (model − satellite, e-h). Regions where the model-satellite difference is smaller than the mean satellite error for a given location (section 2.4) have been shaded in grey.

decreases annual mean TC $C_5H_8$ values by up to $2 \times 10^{16}$ molec. $cm^{-2}$ in northern Australia, the central northern Amazon and southern Africa. TC $C_5H_8$ values are also lower in the NH in JJA, but the zonal mean difference for this region compared to the baseline is about half of that for the subtropical latitudes in DJF (Fig. 4g). On the other hand, zonal mean TC $C_5H_8$ values



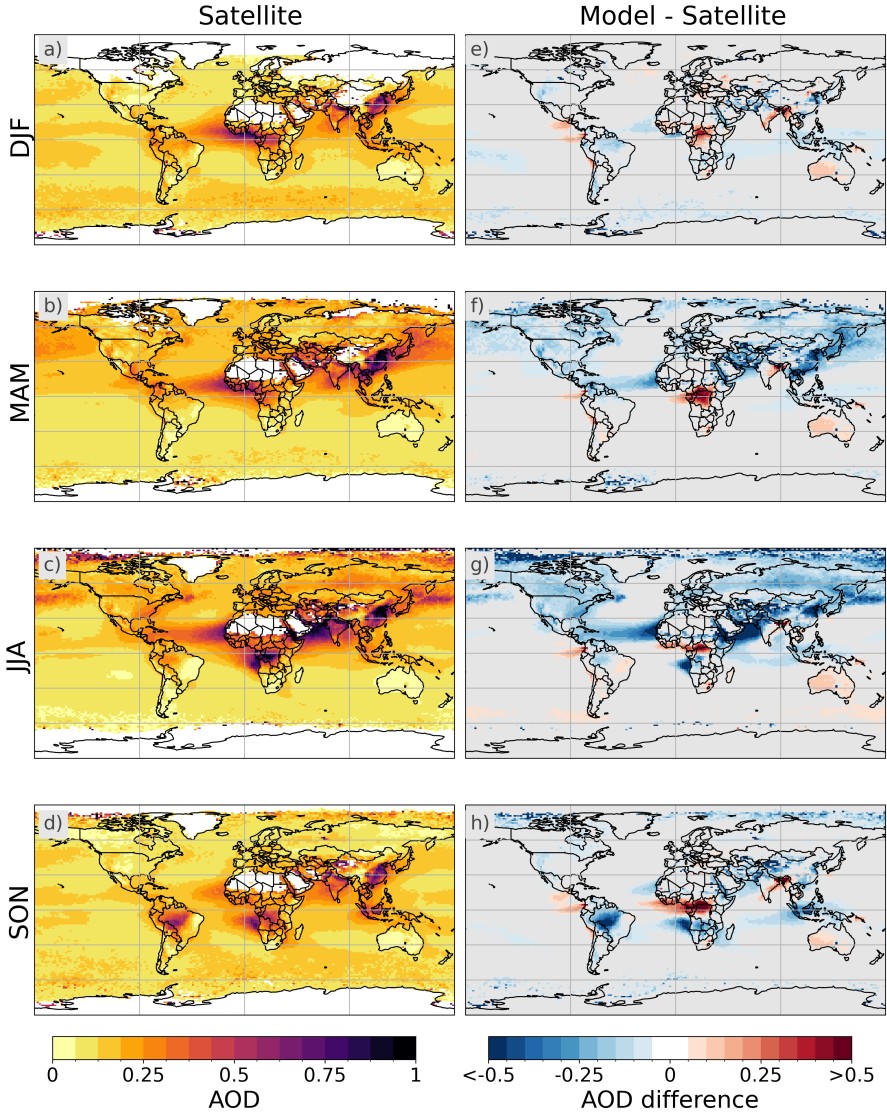

**Figure 3.** Comparison of modelled AOD against satellite AOD observations. The data show the seasonal mean AOD satellite measurements for 2006-2014 (a-d) and the seasonal mean differences between the baseline simulation and satellite observations for the same time period (model − satellite, e-h). Regions where the model-satellite difference is smaller than the satellite error ($\pm(0.05 + 15\%\tau)$) have been shaded in grey.

increase on average by $5 \times 10^{14}$ molec. cm$^{-2}$ (<50%) around the equator in both JJA and DJF, due to greater emissions from tropical forests, particularly in the western Amazon. The HCHO total column (TC HCHO) values tend to respond with changes of a lower relative magnitude but in the same direction as those found for C$_5$H$_8$, except for the equator, where no increase in



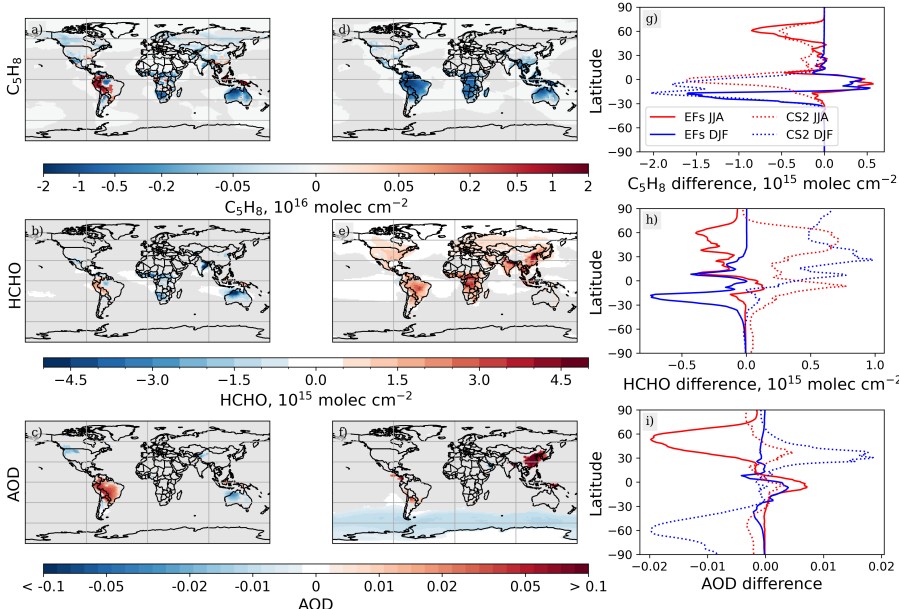

**Figure 4.** Difference from baseline (experiment − baseline) showing impact of updated EFs (a-c, Exp_EF) and the CS2 mechanism (d-f, Exp_CS2) on the annual mean total column isoprene (top row), HCHO (middle row), and AOD (bottom row) for the period 2006-2014. Regions where the difference is not statistically significant at the 95% confidence interval have been masked in grey. The mean difference from the baseline for each latitude for both experiments and two seasons (JJA and DJF) is shown on panels g-i.

TC HCHO occurs in DJF. The lack of response in equatorial TC HCHO (Fig. 4h, between 15° S and 10° N) to the higher $C_5H_8$ emissions suggests another factor is limiting BVOC oxidation at these latitudes. In low-$NO_x$ environments such as the tropical forests, emissions of $C_5H_8$ and monoterpenes, as sinks for the hydroxyl radical (OH), may reduce the atmospheric oxidation capacity (Wang et al., 2024), so the formation of HCHO in the model in these high $C_5H_8$-emitting regions may be oxidant limited.

The impacts of changing the chemistry scheme or reaction rates (Exp_CS2 and Exp_RR) are more uniform over the globe, e.g., Fig. 4d-f for Exp_CS2 (Exp_RR not shown), as they are driven by changes in $C_5H_8$ oxidation and other gas phase reactions that are applied globally in the model. The updated reaction rate constants in the ST chemistry scheme (Table S2) result in a 9% increase in $C_5H_8$ lifetime (Table 2) and greater TC $C_5H_8$ amounts compared to the baseline. In contrast, the CS2 chemistry mechanism leads to a decrease in $C_5H_8$ lifetime by around 50% (Table 2) alongside greater production of HCHO through the more complex oxidation pathways of $C_5H_8$ and monoterpenes (section 2.2). Consequently, TC $C_5H_8$ values decrease by more than $2 \times 10^{15}$ molec. cm$^{-2}$ locally, while TC HCHO increases, particularly in the NH (Fig. 4d,e). These changes are consistent with the findings of Archer-Nicholls et al. (2021), who compared the ST and CS chemistry mechanisms. The decrease in $C_5H_8$ lifetime suggests considerable impacts on the atmospheric oxidising capacity from changing the chemistry mechanism, as found in Weber et al. (2022).





**Table 2.** Global annual mean isoprene lifetime for 2006 and percentage difference between the experiments and the baseline.

| Experiment | Baseline | Ex_RR | Exp_CS2 | Exp_CS2mLC |
|---|---|---|---|---|
| Global mean isoprene lifetime | 7.0 hr | 7.6 hr | 3.4 hr | 3.6 hr |
| Change from baseline | NA | +9% | −51% | −48% |

Figure 5 illustrates the changes in the volume mixing ratios of the oxidants: OH, $O_3$ and hydrogen peroxide ($H_2O_2$) at two altitudes (1 km and 5 km), between the ST and CS2 schemes. There is a clear increase in lower tropospheric OH over regions of substantial biogenic emissions with CS2, e.g. OH volume mixing ratios are over double the baseline values over tropical forests, particularly in DJF (Fig. 5a). This increase is attributed to the addition of $HO_x$ recycling in the more complex chemistry mechanism (Weber et al., 2023). OH at 1 km altitude also increases over some areas of substantial anthropogenic emissions (e.g. Europe and China in DJF), potentially driven by the larger number of alkene reactions included in CS2 (Archer-Nicholls et al., 2021). At 5km, OH mixing ratios increase by a smaller percentage or even decreases when the CS2 mechanism is used (Fig. 5b,d). This change is likely a result of multiple factors, including the greater distance from sources of short-lived trace gases that can form OH (Archer-Nicholls et al., 2021; Weber et al., 2021).

The CS2 mechanism also tends to increase $O_3$ globally, typically by <50% over land at 1 km altitude, and by <30% at 5 km altitude (Fig. 5e-h). Substantial lower tropospheric increases occur over east Asia in DJF, when $O_3$ mixing ratios increase by over 100%. This is likely driven by a more detailed representation of $O_3$ precursors in the CS2 mechanism (see Archer-Nicholls et al. (2021) for a more detailed discussion), including the participation of monoterpenes in gas phase chemistry. The increase at this altitude is even greater for $H_2O_2$, the yield of which increases from $C_5H_8$ reactions with $O_3$, in addition to monoterpene reactions providing a new source of $H_2O_2$ in CS2. In most regions, at both 1km and 5 km, $H_2O_2$ increases by at least 20%. In DJF, the volume mixing ratios are often over 100% greater over land when the CS2 mechanism is used.

Consequently, the two chemistry schemes depict very different relationships between $C_5H_8$ and OH. The significant increase in OH with the inclusion of $HO_x$ recycling in the CS2 mechanism (Exp_CS2) suggests that BVOCs in the standard model using the ST mechanism (baseline) may deplete OH too readily in regions of peak biogenic emissions due to a lack of $HO_x$ recycling. As previously suggested, this impact of BVOCs on oxidant concentrations in ST may explain the lack of a substantial increase in equatorial HCHO in response to higher BVOC emissions at low latitudes in Exp_EF (Fig. 4b,h). The impact of the CS and CS2 chemistry mechanisms on atmospheric oxidants is discussed further in Weber et al. (2021) and Archer-Nicholls et al. (2021).

Finally, changes to both the EFs and chemistry mechanism have implications for aerosols in the model. The updated EFs have a small impact on AOD: values are slightly increased over South America and Borneo (<0.05; <20%), where emissions of monoterpenes from broadleaf forests have been increased, while North America and Australia experience a small decrease in AOD values in JJA (−0.01 to −0.02; −5% to −30%) due to lower monoterpene emissions from needleleaf forest and grass PFTs (Fig. 4c,i). The CS2 mechanism drives significant changes in AOD predominantly away from major BVOC emission



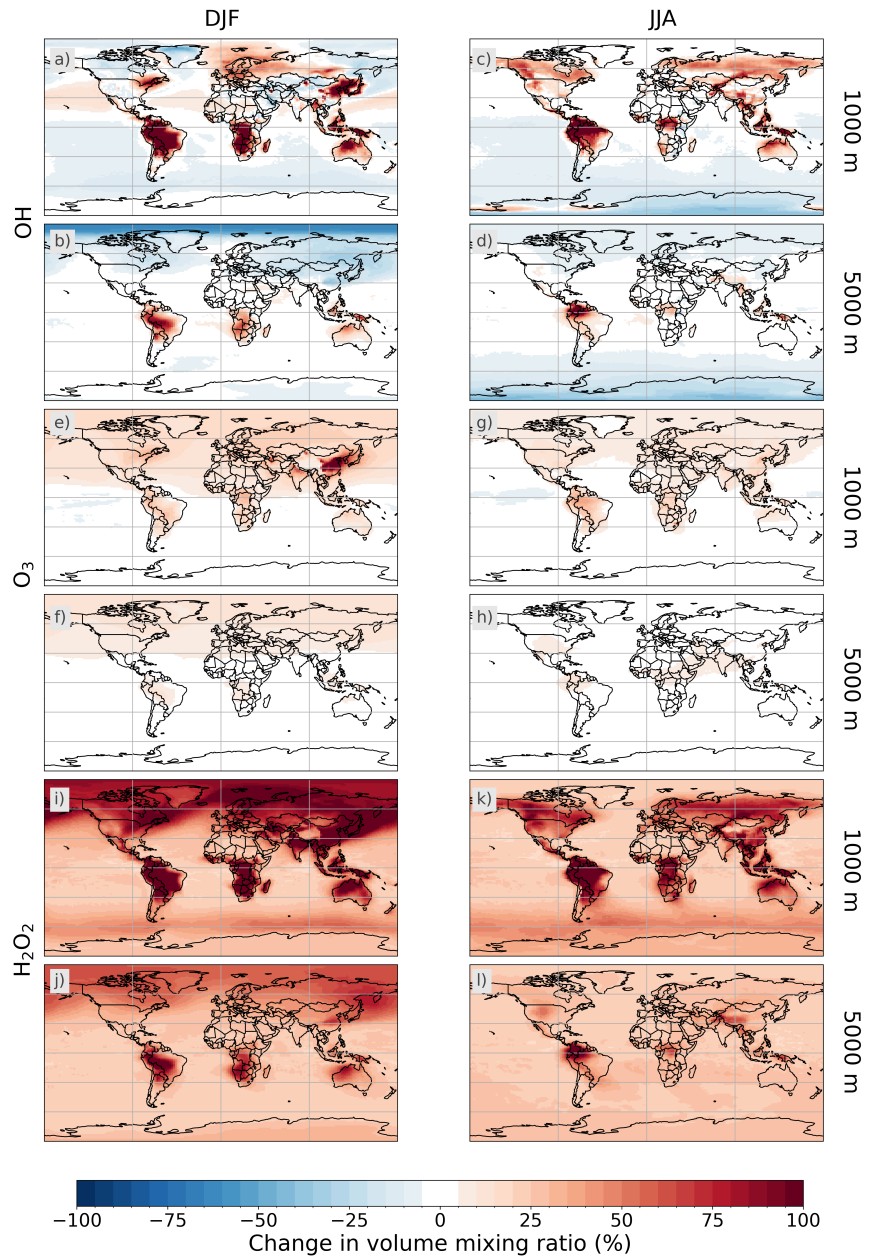

**Figure 5.** Differences in OH (a-d), $O_3$ (e-h) and $H_2O_2$ (i-l) (as % change in average volume mixing ratio for 2006-2014) between experiments using the CS2 and ST mechanisms (Exp_CS2 − baseline) at altitudes of 1 km and 5 km for DJF (left column) and JJA (right column).

sources, such as the tropical forests (Fig. 4f,i); a substantial increase in AOD occurs over China (>0.1), while values decrease over the Southern Ocean (−0.02 at 60° S in DJF). These impacts will be discussed further in section 3.3.





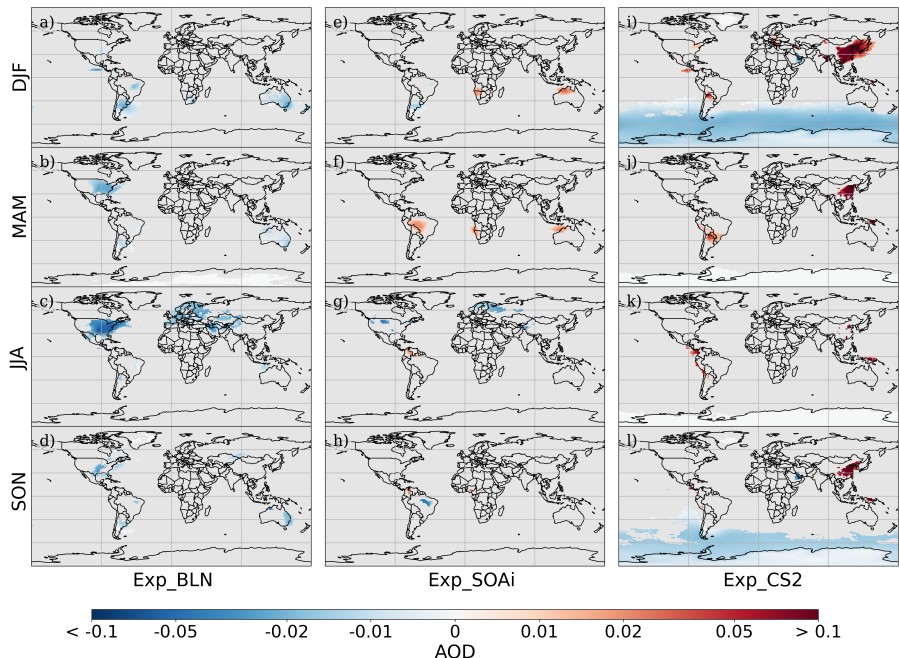

**Figure 6.** Difference from baseline (experiment − baseline) showing impact of including the organically-mediated boundary layer nucleation scheme (Exp_BLN), Sec_Org$_I$ production from isoprene (Exp_SOA$_i$) or the CS2 mechanism (Exp_CS2) on the mean seasonal AOD for the period 2006-2014. Regions where the difference is not statistically significant at the 95% confidence interval have been masked in grey.

## 3.3 Influence of changes on particle formation and growth

Another three experiments focus on processes that directly affect aerosols. The corrected hygroscopicity values have a negligi-
ble impact on AOD regardless of the season (not shown), and do not have a direct connection to $C_5H_8$ or HCHO column values.
Impacts are more substantial from the organically-mediated boundary layer nucleation (BLN), the formation of Sec_Org$_I$ from
$C_5H_8$ and the more complex chemistry mechanism (Fig. 6).

The addition of BLN has the greatest impact in JJA, when AOD values decrease across the eastern US by <0.07, as well
as in Eurasia by 0.01 to 0.05. The magnitude of BLN impacts is smaller in the other seasons, e.g., the significant but small
(<0.02) decrease in Australia in DJF. The organically-mediated nucleation process requires the presence of both Sec_Org$_{MT}$
and $H_2SO_4$, so impacts will be greatest where both are present. Both $H_2SO_4$ and Sec_Org$_{MT}$ are readily available in JJA in
North America. However, the Amazon and other tropical forests, despite being significant sources of BVOCs, are not greatly
impacted by the introduction of BLN, as $H_2SO_4$ production is low in these regions (Fig. 7).

The introduction of BLN may affect the aerosol size distribution, as Sec_Org$_{MT}$ can participate in nucleation when the
mechanism is included, in addition to condensing onto pre-existing particles (section 2.2). This results in increased aerosol
mass in the smaller aerosol modes (nucleation and Aitken), and decreased aerosol mass in the accumulation and coarse modes.
The size of an aerosol particle affects its influence on radiation. The 550 nm AOD calculation in the model is particularly





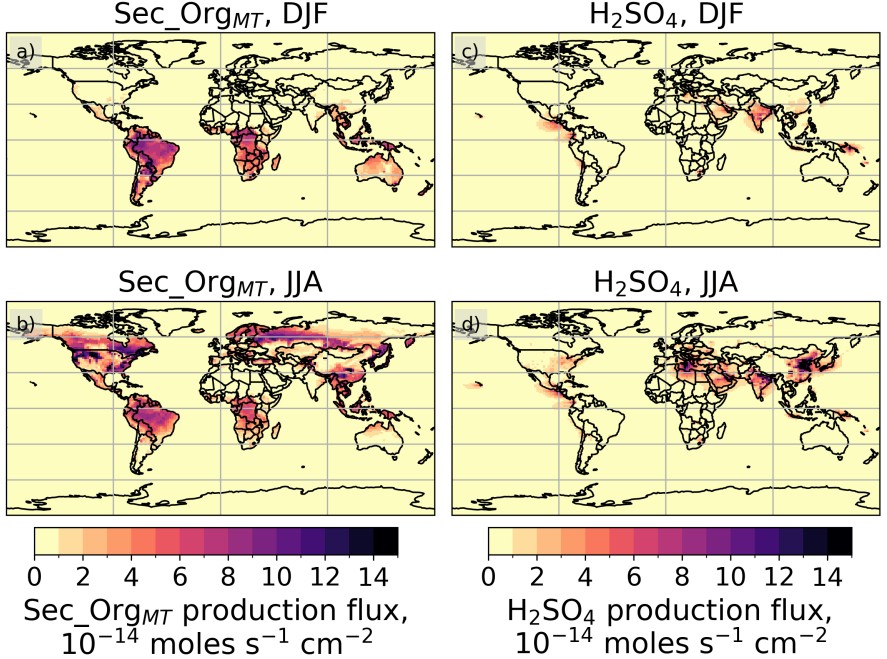

**Figure 7.** Average seasonal fluxes of reactions that result in the production of Sec_Org$_{MT}$ (monoterpenes + OH/O$_3$/NO$_3$, a,b) or H$_2$SO$_4$ (SO$_3$ + H$_2$O, c,d) in moles s$^{-1}$ in the baseline experiment over 2006-2014.

sensitive to aerosol mass in the accumulation mode (see section 3.5). Therefore, a change in the aerosol size distribution, even if the total mass of aerosol does not change substantially, could have significant impacts on the total AOD.

Parts of North America and northern Europe are also significantly affected in JJA (when BVOC emisisons are high in the NH) by the introduction of SOA formation from C$_5$H$_8$ (AOD difference of $-0.01$ to $-0.05$, Fig. 6g). This decrease in AOD is likely due to the removal of the factor of two scaling of Sec_Org$_{MT}$ yield from monoterpene oxidation that occurs when SOA formation from C$_5$H$_8$ is switched on. Monoterpene emissions peak over these regions in JJA (Fig. 8g) and form a substantial proportion of the total BVOC emissions over the boreal forests (Fig. 8k). AOD is slightly, but significantly, increased in MAM

($+0.02$) over the SH (sub)tropical forests and grasses, which are major sources of C$_5$H$_8$ when the original EFs are used, as in this experiment (Exp_SOAi). However, a decrease in AOD values occurs over small parts of South America in DJF and SON (by $-0.02$ to $-0.05$). These different directions of change in South America that depend on the season could be related to the different seasonal cycles of C$_5$H$_8$ and monoterpene emissions and the changes in their relative contribution (Fig. 8i-l). While C$_5$H$_8$ emissions in the Amazon peak over the largest region in DJF and MAM, leading to high Sec_Org$_I$ formation in these

seasons, monoterpene emissions are particularly high in SON, and substantially lower in MAM. Consequently, in MAM the high Sec_Org$_I$ production may be greater than the loss of Sec_Org$_{MT}$ when SOA formation from C$_5$H$_8$ is switched on and the scaling is removed, while in SON the lower contribution from monoterpenes to aerosols may outweigh the addition of Sec_Org$_I$ condensation.




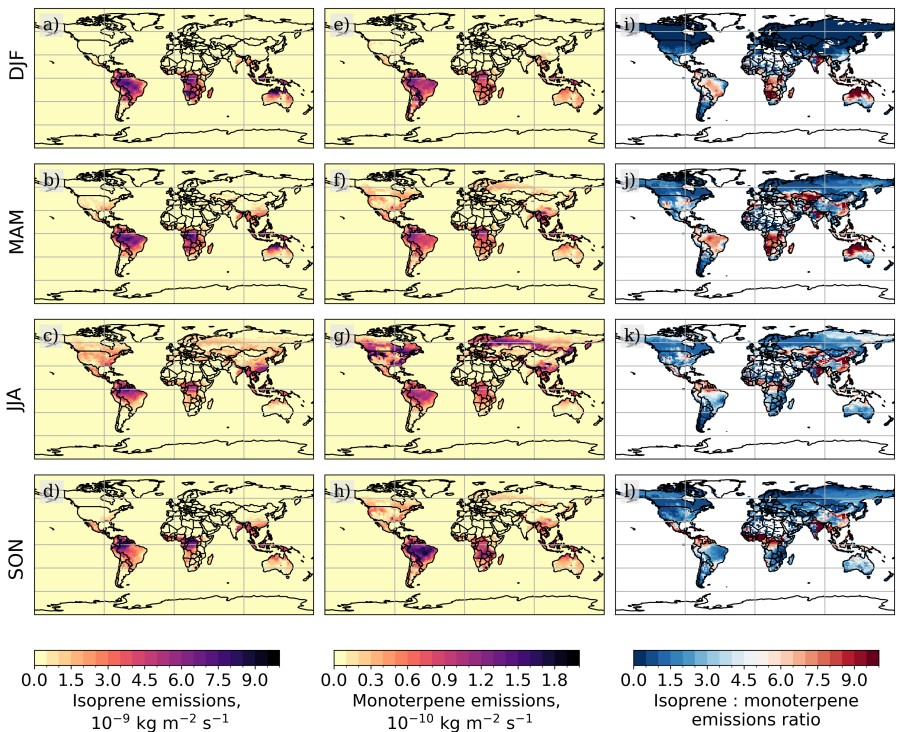

**Figure 8.** Seasonal emissions of isoprene (a - DJF, b - MAM, c - JJA, d - SON) and monoterpenes (e - DJF, f - MAM, g - JJA, h - SON), as well as the isoprene to monoterpene seasonal emission ratio (i-l) in the baseline simulation over 2006-2014.

Finally, the CS2 chemistry scheme drives substantial regional changes in AOD (Fig. 6i-l). AOD values increase over China (>0.1), while the Southern Ocean experiences a widespread decrease in AOD values (−0.02); both changes are most prominent in DJF (Fig. 6i). Figure 9 illustrates the change in total column mass of sulphate, organic matter, black carbon and sea salt aerosols when the CS2 mechanism is used. The sulphate aerosol mass increases by up to 35% over regions associated with high $H_2SO_4$ column values (see Fig. 9 and Fig. 7c,d), but decreases over the oceans and far from anthropogenic emissions, particularly over the Southern Ocean and Antarctica (−50%). In contrast, the changes in mass of other aerosol types are mostly within 5%, with decreases of less than 15% over the oceans and Antarctica for organic matter and black carbon. The AOD and aerosol mass results suggest a change in the amount of sulphate aerosol in the accumulation mode driven by the gas phase chemistry.

We suggest that multiple processes drive these changes in the aerosol size distribution. While the large decrease in sulphate over the Southern Ocean and Antarctica is attributed to the change in dimethyl sulphide chemistry between the two chemistry mechanisms (Fig. S3), other factors dominate elsewhere. In section 3.2, we highlighted how oxidant volume mixing ratios over the lower troposphere significantly increase in response to the more complex chemistry mechanism over regions of high BVOC emissions, as well as in response to some anthropogenic emission hotspots. As a result, the oxidation of aerosol precursors





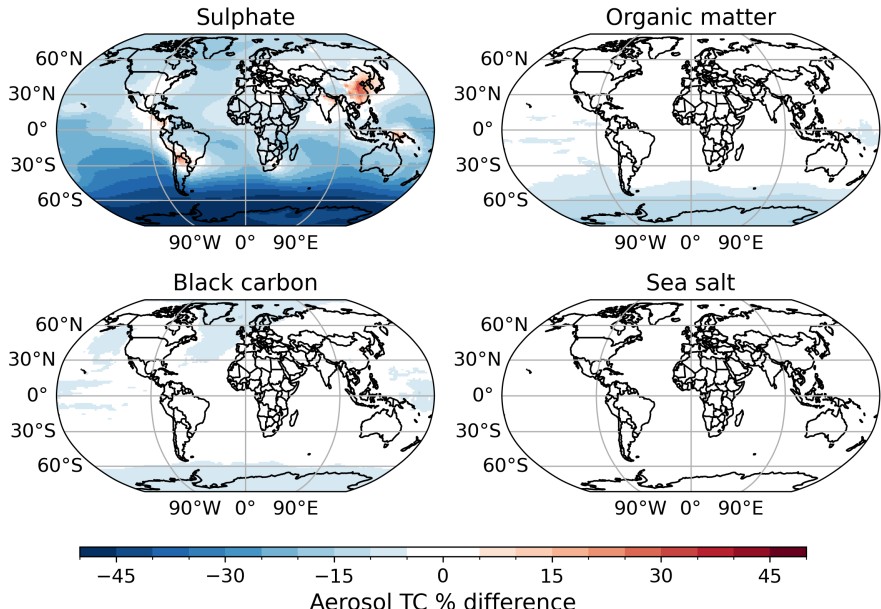

**Figure 9.** Difference in total column aerosol mass between Exp_CS2 and the baseline for 2006-2014. Update colorbar label, add subplot labels

(such as $SO_2$ or $Sec\_Org_{MT}$) is likely to occur closer to source regions in Exp_CS2, consequently changing the location of aerosol formation and growth. Reactions occurring closer to source may be one of the factors driving the regional increases in

sulphate aerosol total column mass (Fig. 9a). While the increased formation of $Sec\_Org_{MT}$ in the boundary layer could result in its greater deposition Weber et al. (2021), the impacts on organic matter seem negligible compared to changes in sulphate aerosol mass.

The change in upper-tropospheric OH will also impact new particle formation. When sulphur dioxide ($SO_2$) reacts with OH in the gas phase, $H_2SO_4$ is formed, which can either form new particles through binary homogeneous nucleation of $H_2SO_4$

and water (Table 3), or condense onto existing aerosols (Mulcahy et al., 2020). This type of nucleation in the model occurs preferentially in the upper troposphere (Fig. S4). However, the low altitude increase in OH mixing ratios is likely to enhance $SO_2$ oxidation at lower altitudes, while lower OH mixing ratios in the upper troposphere will decrease $H_2SO_4$ formation at higher altitudes. Consequently, binary homogeneous nucleation fluxes decrease globally by around 30% when the CS2 mechanism is used (Table 3).

The relative changes in OH, $O_3$ and $H_2O_2$ concentrations will also affect the yields of sulphate aerosol growth pathways. While condensation from gas phase oxidation of sulphur dioxide ($SO_2$) with OH increases globally by 6% (Table 3), the aqueous phase reaction fluxes with $O_3$ and $H_2O_2$ decrease by around 10%, despite the increased volume mixing ratios of $O_3$ and $H_2O_2$ compared to the baseline (Fig. 5). In combination, the shift in the in-cloud and gas phase pathways results in a



**Table 3.** Global and East Asia average percentage change in secondary aerosol production fluxes of sulphate for 2006-2014 between the simulation using the CS2 mechanism (Exp_CS2) and the baseline (Exp_CS2 − baseline).

| Flux | Global change | East Asia change |
|---|---|---|
| Binary homogeneous nucleation | −32% | −24% |
| Aqueous phase reaction with $H_2O_2$ (in-cloud) | −9% | +33% |
| Aqueous phase reaction with $O_3$ (in-cloud) | −12% | +20% |
| Condensation of $H_2SO_4$ | +6% | +17% |

global decrease in overall sulphate aerosol production fluxes of around 1% (all processes shown in Table 3 combined) and,
subsequently, reduces the amount of aerosol in the accumulation mode (Fig. S5).

Although total column sulphate decreases across much of the globe when the CS2 mechanism is used (Fig. 9), some regions experience an increase and, consequently, have substantially higher AOD values, most prominently in parts of east Asia (Fig. 6i). In east Asia, the decrease in binary homogeneous nucleation is of a lower magnitude (−24%, as opposed to −32% globally, Table 3), while the in-cloud processes may increase aerosol mass (change of +30% in the flux for both $H_2O_2$ and $O_3$ reactions
combined, compared to −10% globally). This may be driven by higher $O_3$ and $H_2O_2$ mixing ratios at low and high altitudes (see Fig. 5) and the significant anthropogenic emissions in the region. The emitted aerosol precursors likely have a shorter lifetime due to the high oxidants in the lower troposphere. Consequently, the lower nucleation rates (Table 3) can be compensated for by the greater condensation rates onto sulphate aerosols close to emission sources leading to the positive regional AOD values (Fig 6i-l).b,c

In summary, we find the inclusion of organically-mediated BLN tends to decrease AOD in regions proximate to monoter-penes emissions and $H_2SO_4$ sources, while the inclusion of Sec_Org$_I$ increases AOD values when and where $C_5H_8$ emissions are substantial, but decreases AOD over regions where monoterpenes are the main organic aerosol precursor. Finally, the CS2 mechanism affects AOD by changing oxidant concentrations, leading to the oxidation of aerosol precursors closer to their sources and shifts in the sulphate particle formation and growth pathways. The CS2 mechanism results show the importance
of the connections between gas-phase chemistry and the aerosol size distribution, as previously highlighted by Karset et al. (2018) and O'Connor et al. (2022).

### 3.4 Impact of land cover

TC $C_5H_8$, TC HCHO and AOD are expected to respond to changes in the land cover distribution, which is relevant to the calculation of BVOC emissions. Here, the ESA CCI land cover data is used, similarly to that employed in an evaluation
study of UKESM1 (Sellar et al., 2019), to investigate the impact of using a more realistic PFT distribution on atmospheric composition. Key changes in the land cover when implementing the ESA dataset include the decrease in broadleaf forest cover in eastern South America and south of the Congo, the expansion of needleleaf forest at the cost of grasses and shrubs in Siberia,



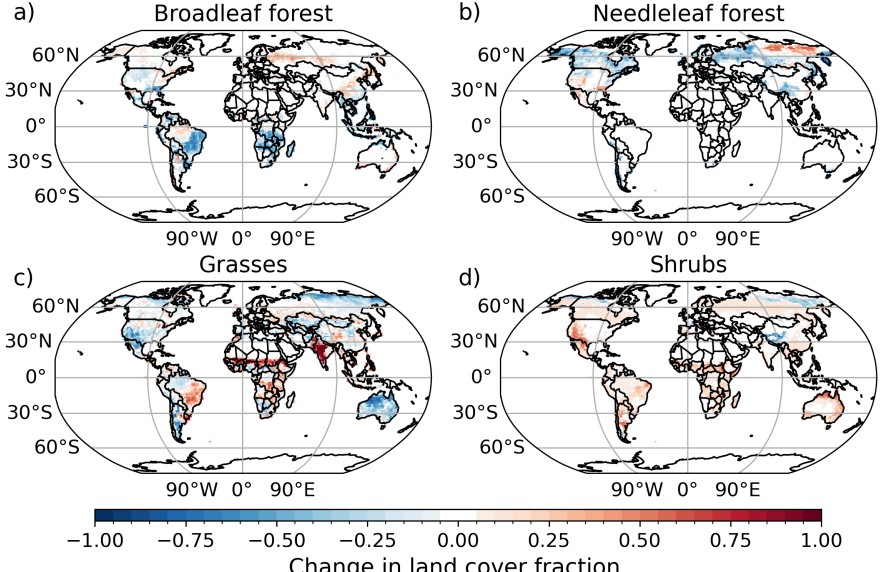

**Figure 10.** Change in grouped PFT fractions between land cover derived from a free running simulation and the ESA CCI land cover ancillary. Broadleaf forest (a) includes broadleaf deciduous, evergreen tropical and evergreen temperate trees, needleleaf forest; (b) combines needleleaf deciduous and evergreen trees; grasses (c) merges the C3 and C4 grass, crop and pasture PFTs; while shrubs (d) refers to both deciduous and evergreen shrubs.

the loss of grassland in Australia, and the expansion of grasses south of the Sahel and in western India, as well as shrubs in southwestern North America (Fig. 10). The subsequent changes in BVOC emissions will therefore vary regionally, following this wide range of land cover changes. In addition, the ESA CCI land cover does not separate the grasslands into grasses, crops and pastures as in the UKESM1.1-derived land cover, but groups all of these PFTs into the C3 grass and C4 grass categories. These grass PFTs have higher emission factors for both $C_5H_8$ and monoterpenes than the crop and pasture PFTs. Therefore, the use of the observation-based land cover not only affects the spatial distribution of the PFTs, but also results in the complete loss of crops and pastures in favour of high emitting grasses. The impacts of land cover change become more complex when the new EFs are also included, as the EFs may enhance or oppose the change in atmospheric composition due to land cover spatial distribution changes alone.

Figure 11 illustrates the changes in TC $C_5H_8$, TC HCHO and AOD for DJF and JJA when the ESA CCI land cover with the original or new EFs is used. In all cases, there is a substantial decrease in AOD over the savannas south of the Sahel and over India (by over 0.1, or around 25%), which is driven by the presence of grasses over previously barren land limiting dust emissions (not shown). Both experiments are also characterised by significant decreases in TC $C_5H_8$ ($-0.4$ to $-2 \times 10^{16}$ molec. cm$^{-2}$) and TC HCHO (within $-5 \times 10^{15}$ molec. cm$^{-2}$ in DJF) in NW Australia, which is driven by the loss of grassland and enhanced further by lower EFs for grass PFTs when the new EFs are included (Fig. 11a,b,d,e). Although the African savannas



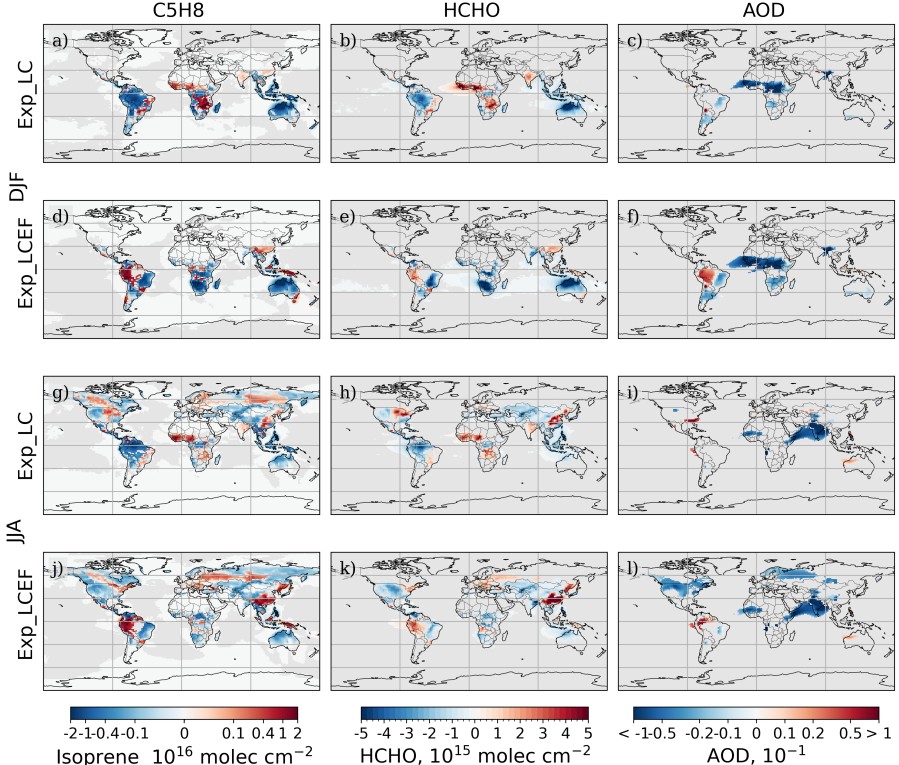

**Figure 11.** Difference from baseline (experiment− baseline) showing the impact of using the ESA CCI derived land cover ancillary with or without the updated emission factors (Exp_LCEF and Exp_LC, respectively) on the mean DJF and JJA total column isoprene and formaldehyde, and AOD for the period 2006-2014. Regions where the difference is not statistically significant at the 95% confidence interval have been masked in grey.

see a significant increase in TC $C_5H_8$ and TC HCHO values when only the land cover is changed, the lower grassland emissions with the new EFs dampen the response for these trace gases in the region.

There are large, but varied, changes in trace gas columns over the Amazon. Isolating the land cover change leads to a significant decrease in both trace gases in the west by $< 50\%$ for $C_5H_8$ and by $< 25\%$ for HCHO, but an increase in the east in both seasons ($< 100\%$ for $C_5H_8$, $< 20\%$ for HCHO). In this region, the use of updated EFs leads to changes that are opposite in sign but of similar magnitudes. The direction of the changes is also opposite for central Africa when the new EFs are used (compare Fig. 11a and 11b). In both South America and Africa, where grasslands are expanded in the ESA CCI LC compared

to the model-derived land cover, the inclusion of new EFs alongside the observational land cover greatly changes the response; the new grassland EF is very low leading to substantially decreased $C_5H_8$ emissions in these regions (Weber et al., 2023). The impact of the new PFTs is also clear in South America in DJF for AOD (compare Fig. 11c and 11f). The increased emissions



of monoterpenes from broadleaf forests in the central and western Amazon lead to greater production of Sec_Org$_{MT}$ in this region and increase aerosol mass (Fig. 11f,l).

Particularly in JJA, there are more complex changes in mid- to high-latitude Eurasia, as broadleaf forests expand in the west, and grasslands decrease in favour of needleleaf forests in the east. Consequently, TC C$_5$H$_8$ increases in the west (for both land cover experiments) by $> 75\%$, but decreases in the far northeast ($-50\%$ to $-90\%$). The land cover change tends to decrease BVOCs in western North America, where grasses and broadleaf forests are partially replaced by shrubs, though differences remain within $1 \times 10^{16}$ molec. cm$^{-2}$. In general, TC HCHO changes follow those of TC C$_5$H$_8$, while AOD values decrease in

the northern high latitudes in JJA when both the alternative land cover and new EFs are combined, as both the fractional cover and monoterpene EFs decrease for needleleaf trees.

     In summary, globally, the land cover and EFs significantly change the spatial distribution and total amount of BVOCs emitted, therefore modifying the distribution of HCHO and SOA precursors and, subsequently, organic aerosol and AOD, as seen over the Amazon. The land cover additionally changes dust emissions, which can have significant impacts on AOD.

## 470    3.5    Combination of model updates

In this section we focus on the interactions between the studied processes under three combination scenarios. The first merged experiment includes the new EFs; updated reaction rates and hygroscopicity values; the organically-mediated BLN and formation of Sec_Org$_I$ from isoprene (Exp_STm). The other two simulations investigate the impact of using the more complex CS2 chemistry scheme alongside other updates, with or without the observation-based land cover (Exp_CS2mLC and Exp_CS2m,

respectively). In addition to studying the impacts on TC C$_5$H$_8$, TC HCHO and AOD, we investigate how the various processes impact aerosol mass and number in different modes.

     Significant differences (relative to the baseline simulation) in TC C$_5$H$_8$, TC HCHO and AOD occur with the updated chemistry mechanism when all other processes but the ESA CCI landcover are included (Fig. 12, compare Exp_STm (top row) and Exp_CS2m (middle row)). With the standard chemistry mechanism, the new EFs, alongside reaction rate updates, drive

tropical TC C$_5$H$_8$ increases of 0.4 to $2 \times 10^{16}$ molec. cm$^{-2}$ (25 to 100%), while the more complex CS2 mechanism counteracts these changes and results in a substantial decrease in TC C$_5$H$_8$ ($-0.5$ to $-2 \times 10^{16}$ molec. cm$^{-2}$) over tropical BVOC emission sources (see section 3.2 for discussion of individual processes). Similarly, TC HCHO has the strongest response to the choice of chemistry mechanism (compare Fig. 12d and e), resulting in higher TC HCHO values in most regions of the world ($< 5 \times 10^{15}$ molec. cm$^{-2}$). The new EFs do result in lower TC HCHO regardless of chemistry mechanism in some regions, notably

in northern Australia and southwestern Africa (cf Fig. 4e and Fig. 12e). The alternative land cover has further minor regional impacts on both traces gases, e.g. over China and eastern South America, following the transitions described in section 3.4. The combination of all processes (Exp_CS2mLC) also results in a substantial decrease in C$_5$H$_8$ lifetime compared to the baseline ($-48$ %, section 3.2, Table 2). Adding the other processes to the more complex chemistry scheme results in a minor change in C$_5$H$_8$ lifetime (3.6 hr for Exp_CS2mLC compared to 3.4 hr), suggesting the CS2 mechanism is the major driver of this signal.

Overall, the choice of chemistry scheme dominates TC C$_5$H$_8$ and TC HCHO impacts when combined with the other studied processes, with regional changes driven by EF and land cover updates.





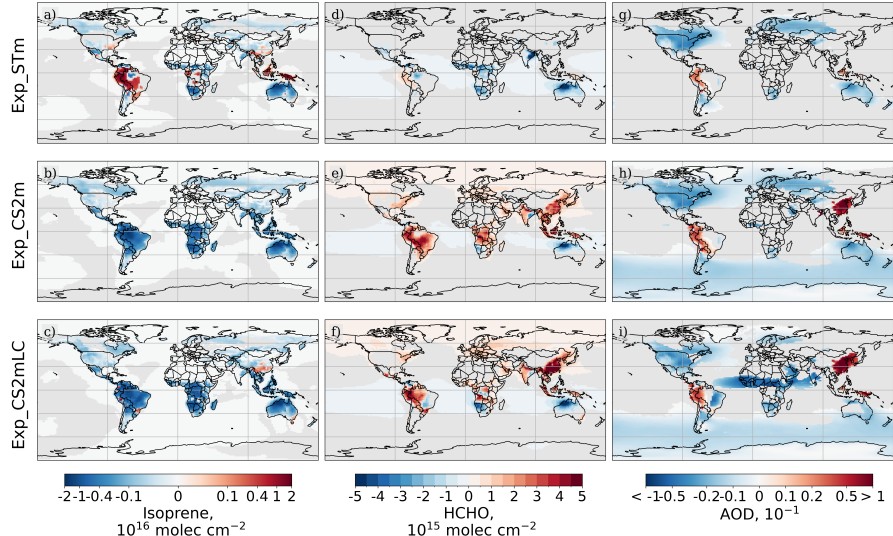

**Figure 12.** Annual mean differences in the total columns of isoprene (a-c) and HCHO (d-f), and AOD (g-i), between the merged experiments and the baseline for 2006-2014. Regions where the difference is not statistically significant at the 95% confidence interval have been masked in grey.

There are similarities in the AOD results for all three merged experiments. North American and Eurasian AOD values are decreased by < 0.05 compared to the baseline (Fig. 12g-i). This is consistent with the lower JJA AOD in the NH when either BLN or $Sec\_Org_I$ are included in the model (section 3.3). The key impact of using the CS2 mechanism alongside other processes is the increase in AOD by more than 0.1 over China, as well as the lower AOD over the ocean in the SH (c.f Fig. 12h,i and Fig. 6i-l in section 3.3). The land cover change results in reduced AOD values over eastern South America, south of the Sahel and India (section 3.4). Additionally, Eurasia is not as strongly affected by the NH decrease when the ESA CCI land cover is used, likely due to the broadleaf forest expansion limiting the impacts of BLN and new EFs.

Figure 13 illustrates that, when all the processes are combined, the greatest differences in AOD occur in the accumulation mode. Particularly visible are the continental decrease in AOD (<0.06), as well as the >0.1 increase over China (Fig. 13f). The CS2 mechanism leads to a global decrease in sulphate aerosol mass (seciton 3.3), which affects the accumulation mode, whilst BLN and SOA formation from $C_5H_8$ tend to shift aerosol mass towards the smaller modes (Fig. S6). Mineral dust AOD values decrease in the Sahel and India, and increase over Australia, all within 0.06, highlighting the influence of land cover change on dust emissions in these regions.

Our experiments highlight how processes involved in the biosphere impacts on atmospheric chemistry and aerosols have a wide range of effects on the aerosol size distribution and are not limited to changes in secondary organic aerosol formation. Further, the processes may affect each other, resulting in non-linear effects when they are combined.



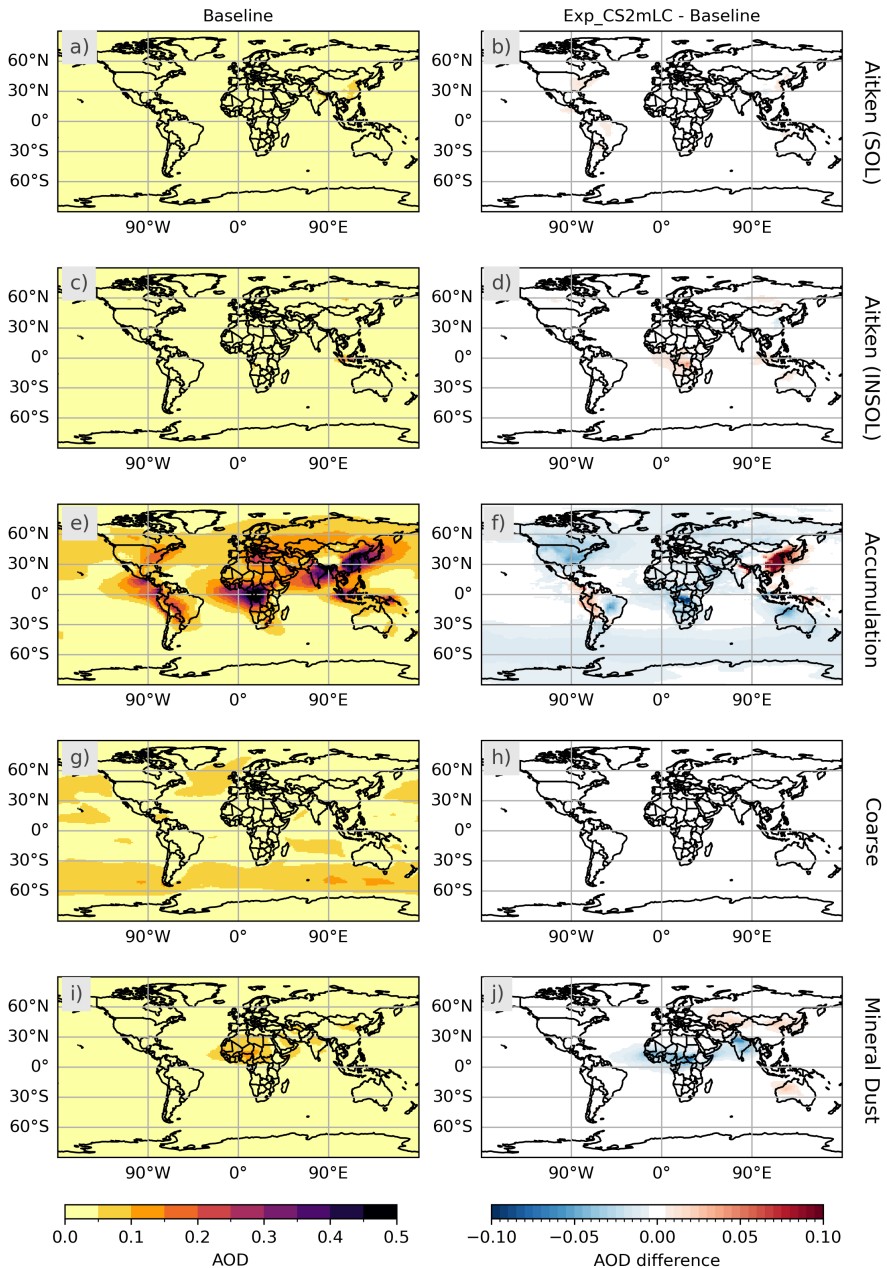

**Figure 13.** Baseline AOD (left column) and the difference in AOD between the CS2 merged experiment with ESA CCI LC (Exp_CS2mLC) and the baseline simulation (right column) for different aerosol modes (from top to bottom: Aitken sol, Aitken insol, accumulation, coarse, mineral dust).





## 3.6 Impact of model processes on satellite-model biases

The initial satellite-model comparison identified a mix of positive and negative regional biases in the TC $C_5H_8$ and AOD, while
TrC HCHO tended to be underestimated by the model (section 3.1). In this section we outline how the biosphere-atmosphere processes impact these biases by comparing the differences between the simulation inclusive of all processes (Exp_CS2mLC) and the satellite data, and the original differences found for UKESM1.1 (section 3.1). The model data are profiled in time and space to match the observations as closely as possible and averaging kernels are applied to the model HCHO (section 2.4.1).

Figure 14 shows the new model bias as a percentage of the original bias identified in section 3.1. The significant decrease
in TC $C_5H_8$ values in the tropics (discussed in section 3.2 and 3.5) results in lower $C_5H_8$ bias values in these regions in DJF and MAM (<25% of the original bias magnitude, Fig. 14a-b). However, the change in TC values is so large that in DJF the TC $C_5H_8$ values are now too low in some regions (e.g. values $<-100\%$ in the Congo and eastern South America, Fig. 14a). Further, the magnitude of the negative bias calculated for JJA and SON over eastern South America (section 3.1) is increased when all the processes are combined. Overall, the new process representation, alongside the more complex chemistry scheme
and observational land cover, improve the model agreement with the satellite observations over $C_5H_8$ emission hotspots, particularly in MAM. However, challenges remain, such as negative biases becoming more prevalent for parts of eastern South America. The magnitude of the global mean difference between the simulated and satellite-observed TC $C_5H_8$ decreases by 50%, suggesting the improvements in the tropics have a substantial impact on the global average bias.

The substantial increase in HCHO driven by the more complex chemistry mechanism (sections 3.2 and 3.5) leads to some
regional improvements in model performance. The bias values decrease by 5 to 50% in the central and western Amazon in JJA and SON (Fig. 14g,h) and over China in all seasons (Fig. 14e-h). However, the bias increases by over 50% in southern Africa and northern Australia in DJF and MAM (Fig. 14e,f) in response to updated EFs and land cover (sections 3.4 and 3.5). The relative change in HCHO model-satellite differences is smaller than the change in TC HCHO values between experiments, which may be due to the sensitivity of the satellite retrieval to different altitudes (Supplement 9). Consequently, the magnitude
of the global mean model-satellite difference in the TrC HCHO increases slightly (<3%), despite the regional improvements.

The effects of the studied process on AOD are regionally mixed and seasonally dependent (sections 3.2-3.5) and, consequently, the process impacts on the biases are varied. None of the processes ameliorate the negative bias over North America in JJA (Fig. 14k). The altered process impacts are more notable elsewhere. In JJA and SON, the AOD bias decreases by up to 50% over the central Atlantic Ocean, likely in response to land cover change in Africa (section 3.4). There is also a slight increase
in AOD values in the west of the Amazon, where the bias is reduced by at least 5% throughout the year. This is predominantly driven by changes in land cover and EF updates (section 3.4). Over China, the AOD bias magnitude decreases and, in some parts, changes sign so that AOD values are now overestimated in response to the chemistry-mechanism-driven increase in AOD (section 3.3). As the AOD bias in the baseline model was negative and the combination of biosphere-atmosphere processes decreases AOD values across much of the globe, the magnitude of the average global AOD bias increases by around 0.01
(<26%), despite the substantial regional improvements.





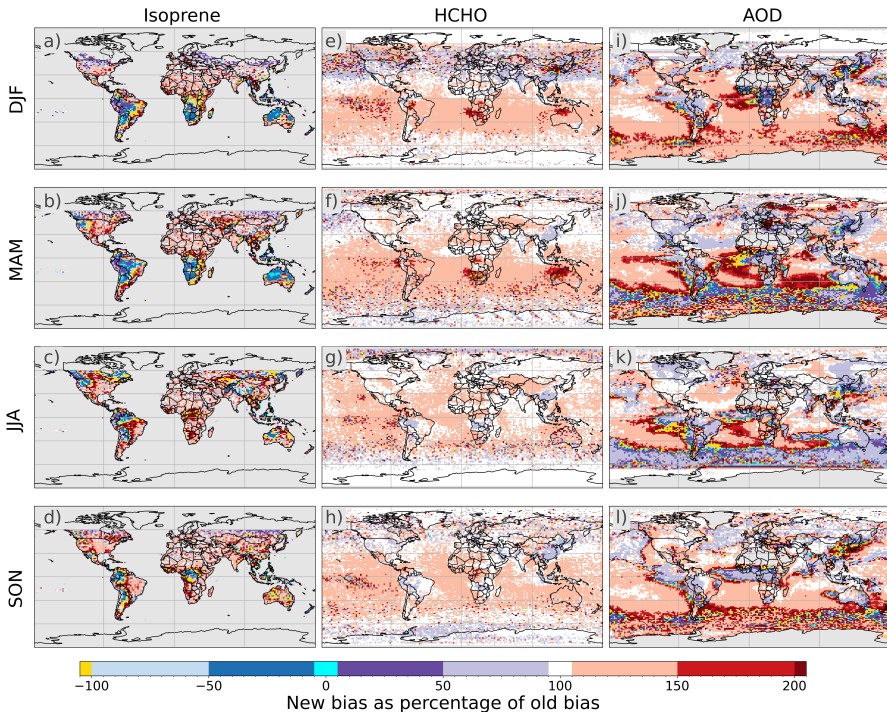

**Figure 14.** Satellite-model bias for Exp_CS2mLC as a percentage of the baseline model bias. 0 indicates complete 'mitigation' of the bias. Negative values indicate a change in the sign of the bias. Values over 100 (or less than−100) show the bias has increased in magnitude. The grey colour indicates no satellite data was available for a given location. Left column – isoprene, middle – HCHO, right – AOD.

## 3.7 Implications for aerosol radiative effects

Lastly, we explore the aerosol-radiation interactions in UKESM1.1 and how they are affected by the biosphere-atmosphere processes. The aerosol-radiation interactions are evaluated by calculating the direct radiative effect (DRE) defined as the difference in net top of atmosphere (TOA) radiative flux due to aerosol scattering and absorption for both shortwave and long wave components following Ghan (2013) (section 2.4.2). In this case, a negative DRE indicates greater outgoing radiation at the TOA.

Figure 15a,b shows the DRE in the baseline for shortwave (SW) and longwave (LW) radiation. We calculate a net global mean DRE (SW+LW) of $-1.73 \pm 0.001$ W m$^{-2}$ (the standard error is used to represent the uncertainty). Aerosol-radiation interactions have more impact in the SW (global mean of $-2.08 \pm 0.001$ W m$^{-2}$); there is a predominantly cooling effect up to $-4.0$ W m$^{-2}$ over land and greater over parts of the oceans (Fig. 15a). Smaller areas with positive effects (typically $< +3.0$ W m$^{-2}$) occur over the Sahara, the Arabian Peninsula and central China. The SW DRE values are also positive over the Atlantic west of southern Africa, likely due to the presence of black carbon from biomass burning. The LW DRE is on average of a




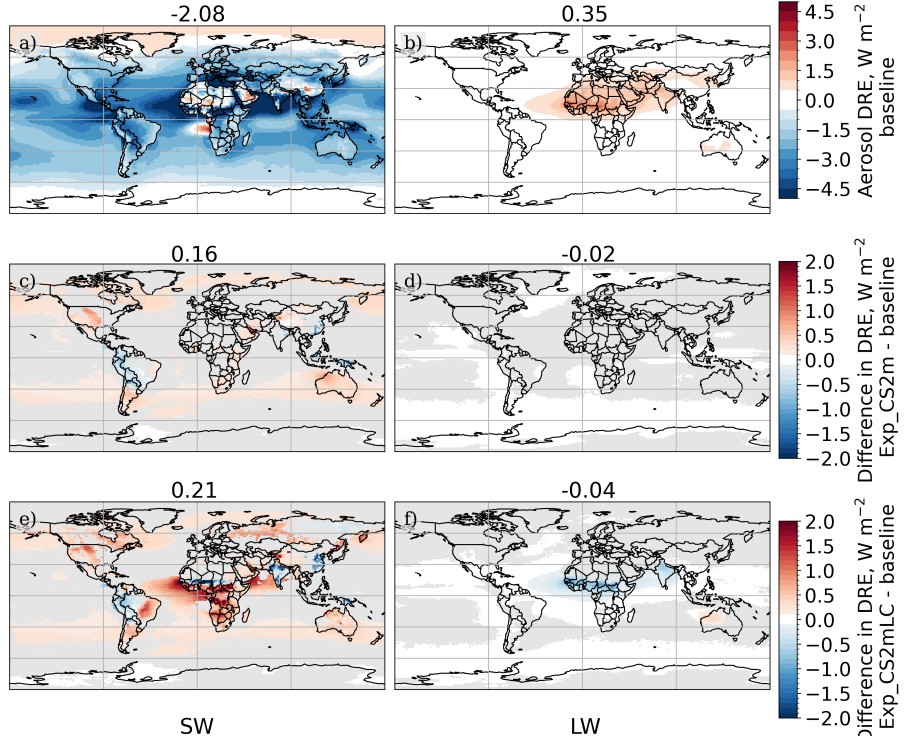

**Figure 15.** Temporal mean (2006-2014) aerosol direct radiative effect (DRE, see Section 2.4.2) in the baseline (a: shortwave, b: longwave), the difference in DRE driven by the biosphere-atmosphere processes excluding land cover (Exp_CS2m − baseline, c: shortwave, d: longwave) and the difference in DRE due to the biosphere-atmosphere processes including land cover (Exp_CS2mLC – baseline, e: shortwave, f: longwave). Numbers over the panels show the global average for the given subplot. Regions where the difference is not statistically significant at the 95% confidence interval have been masked in grey on panels c-f.

lower magnitude ($+0.35 \pm 0.0002$ W m$^{-2}$ global mean), although a warming signal ($< +3.0$ W m$^{-2}$) is found over dust source regions, such as the Sahara and the Arabian Peninsula (Fig. 15b).

The impact of including all processes except for the land cover is shown on Fig. 15c,d; this is the difference in DRE between Exp_CSm and the baseline. There is a statistically significant (95% confidence) change between +0.2 to +0.4 W m$^{-2}$ in the SW component over high latitude oceans in both hemispheres, likely driven by the change in DMS chemistry scheme (section 3.3). Positive differences also occur over northwestern Australia ($< +0.8$ W m$^{-2}$) and the central USA ($< 1.0$ W m$^{-2}$). All of these are locations where AOD decreases when the new processes are included (section 3.5). Subsequently, the scattering of radiation by aerosols is reduced, leading to the warming effect in the SW. The opposite occurs in small regions in western

South America, China and parts of southeast Asia. Differences of around $-0.6$ W m$^{-2}$ to $-1.0$ W m$^{-2}$ occur for the SW DRE in these regions, which are also characterised by higher AOD values compared to the baseline (section 3.5). The DRE in the




LW remains largely unaffected (Fig. 15d). Globally, the processes have a net (SW + LW) effect of + 0.14 ± 0.001 W m$^{-2}$ on the DRE and reduce the magnitude of the cooling from aerosol-radiation interactions compared to the baseline.

The inclusion of the observational land cover, alongside all other processes, increases the net global mean DRE by +0.17 ± 0.001 W m$^{-2}$ (Exp_CS2mLC − baseline) through a combination of the changes in aerosols and differences in the land surface albedo. In addition to changes described for Exp_CS2m above, positive differences in the SW are found in central and southern Africa and their outflow regions over the Atlantic (> +1.8 W m$^{-2}$ regionally, Fig. 15e), alongside smaller magnitude impacts (< +1.6 W m$^{-2}$) in much of eastern South America, North America and western Russia. Negative SW effects are strengthened in

China and also found in northern India and on the southern boundary of the Sahel. These changes in the SW DRE correspond to previously discussed differences in AOD (section 3.5). Further, significant impacts occur for the LW DRE in regions where using the observational land cover changes dust emissions compared to the baseline (Fig. 15f, section 3.4). Values increase in northwestern Australia (< +0.6 W m$^{-2}$), where dust emissions have increased, and decrease south of the Sahel and in northern India (<−1.2 W m$^{-2}$), which are characterised by lower dust emissions.

The changes in the present-day global aerosol DRE, due to the impacts of biosphere-atmosphere processes, calculated here (+0.14 W m$^{-2}$ and +0.17 W m$^{-2}$) are greater than the radiative forcing from aerosol-radiation interactions associated with pre-industrial to present-day land cover changes and their impacts on BVOC emissions as found in previous studies. For example, D'Andrea et al. (2015) find an aerosol direct effect of +0.05 to +0.129 W m$^{-2}$ when comparing simulations with BVOC emissions for the years 1000 and 2000, while Heald and Geddes (2016) calculate a direct radiative forcing of +0.017 for

biogenic SOA associated with land use change between 1850 and 2000. The uncertainty due to process representation identified in this study will impact the pre-industrial background aerosol state and, consequently, will affect estimates of pre-industrial to present-day anthropogenic aerosol (Carslaw et al., 2017) and land use forcing.

## 4   Conclusions

This research identifies satellite-model biases of relevance to biosphere impacts on atmospheric chemistry and aerosols and

quantifies their sensitivity to new process representation in the model. We find that in the standard UKESM1.1 model setup, TC C$_5$H$_8$ is overestimated by a factor of 5 in the tropics in DJF and MAM, while smaller magnitude negative biases occur regionally. The seasonality in South America is not well captured, as the sign of the bias changes seasonally. Further, where C$_5$H$_8$ values are underestimated in JJA and SON, a negative bias (<80%) is found for TrC HCHO. This suggests that the underestimate of C$_5$H$_8$ impacts its oxidation products, including HCHO. In SON, there is also a negative bias in AOD over the

central Amazon (60 to 100%). Elsewhere, AOD values are underestimated over land in the northern hemisphere, but positive biases are found in the Congo and western Australia.

     The use of updated EFs decreases TC C$_5$H$_8$ over (sub-)tropical grasslands, but increases TC C$_5$H$_8$ values over the tropical forests. Global impacts occur for the more complex representation of BVOC oxidation in the CS2 mechanism, which decreases TC C$_5$H$_8$ through a 50% reduction in the mean C$_5$H$_8$ lifetime. This global decrease results in better agreement between the



satellite observations and UKESM1.1 in the western part of the Amazon, but strengthens negative biases in the east of South
       America.

       The same more detailed chemistry in the CS2 mechanism leads to an increase in global HCHO, which may counter some of
       the negative TrC HCHO biases. The more complex BVOC oxidation with $HO_x$ recycling alongside HCHO formation from the
       oxidation of monoterpenes increase TC HCHO without any concurrent changes to $C_5H_8$ emissions. This result highlights that
there may be missing sources of HCHO in the standard model set up. Considering the positive TC $C_5H_8$ biases in many tropical
       regions, these could be from other VOCs, whether biogenic or anthropogenic, or the HCHO yield from BVOC oxidation
       may be underestimated. This is an important consideration when using HCHO products as a proxy for BVOCs, as HCHO
       satellite-model bias results based on the standard model set up would suggest BVOCs were underestimated in the model, while
       the satellite-model comparisons of TC $C_5H_8$ show significant regional overestimation. The impact of the changes in process
representation on the magnitude of the TrC HCHO bias is limited by the low sensitivity of the satellite retrieval to the lower
       troposphere, where the greatest differences between model experiments occur in high BVOC-emitting regions.

       Impacts of the CS2 scheme are also significant for aerosol loading. AOD increases in China by up to 50% of the annual
       average when the CS2 mechanism is used. This is attributed to the shorter lifetime of aerosol precursors due to higher oxidant
       concentrations, leading to higher aerosol loading in source proximate regions, alongside shifts in the aerosol size distribution
driven by decreased nucleation and increased condensation onto sulphate aerosols.

       Some regional biases are reduced by changes to BVOC and dust emissions through the implementation of new EFs and
       observationally-derived land cover. South America is particularly affected, as BVOC emissions from grasslands are reduced,
       leading to significant decreases in $C_5H_8$, HCHO and AOD in the east, while higher forest cover and an increase in forest EFs
       in the west results in higher values for both trace gases and AOD.

None of the processes studied here improve the negative NH AOD bias when compared to MODIS AOD. An alternative
       set up of the GLOMAP model allows for the inclusion of ammonium, nitrate and sodium nitrate aerosols and could increase
       aerosol mass and the simulated AOD value (Jones et al., 2021). Further experiments could investigate how the inclusion of
       nitrate aerosols would modify the processes studied here.

       The combination of all the processes leads to a decrease in the aerosol mass in the accumulation mode in favour of greater
aerosol mass in the Aitken modes. It is the decrease in the accumulation mode that dominates the AOD response. These changes
       in AOD are reflected in significant differences in the aerosol DRE when all biosphere-atmosphere processes are included. For
       example, the higher AOD values over China are associated with an increase of the magnitude of the negative DRE in the region,
       while the global decrease in aerosol loading leads to a positive shift in the net global DRE by +0.17 W m$^{-2}$.

       In conclusion, the satellite model comparison has identified strong regional variation in satellite-model derived biases for
TC $C_5H_8$ and AOD, while TrC HCHO values are generally underestimated globally. Considering the opposite sign of the $C_5H_8$
       and HCHO biases in the baseline, a more complex simulation of BVOC oxidation with the inclusion of $HO_x$ recycling and
       the formation of HCHO during monoterpene oxidation included in the CS2 chemistry mechanism is necessary to simulate
       BVOC-HCHO relationships more accurately. Furthermore, additional work is required to improve the representation of the
       seasonal cycle of $C_5H_8$ emissions in South America, especially considering that the Amazon forest is a globally-significant



BVOC source. Negative biases in AOD that are not reduced by the inclusion of the processes studied could suggest a missing source of aerosol in the model, e.g. nitrates and ammonium. The greatest impacts on both trace gases and AOD occur when the chemistry mechanism is changed. The strong response of the aerosol size distribution to the chemistry mechanism highlights the sensitivity of aerosols in the UKESM model to precursor gas-phase chemistry. Careful consideration of process representation in the model is required for studies investigating BVOC forcings, as the inclusion of new processes significantly affects present-

day aerosol radiative effects.

*Data availability.* The isoprene data are available at https://doi.org/10.1029/2021JD036181 (Wells et al., 2022) on request from the corresponding author of this article. Level 2 OMI HCHO data files are available on request in HDF5 format from https://h2co.aeronomie.be/. The level 2 AOD data (MYD04_L2, Levy et al. (2017)) is available on the Centre for Environmental Data Analysis (CEDA) archive on Jasmin (Lawrence et al., 2013; National Aeronautics and Space Administration, 2021).

*Author contributions.* RP, RMD, FMO'C, and ES designed the research study. ES and RP analysed the satellite data. ES ran the simulations with support and advice from all coauthors. DG prepared the baseline simulation and advised on implementation of aerosol processes. JW prepared code for the implementation of CRI-Strat 2 and ran supplementary simulations. ES prepared the manuscript with scientific, editorial and technical input from all coauthors.

*Competing interests.* No competing interests

*Acknowledgements.* This research and ES has been supported by the Natural Environment Research Council (NERC) through a SENSE CDT studentship (grant no. NE/T00939X/1) and a CASE award from the UK Met Office. FMO'C was supported by the Met Office Hadley Centre Climate Programme funded by DSIT and the European Union Horizon 2020 Research Programme ESM2025 (grant agreement number 101003536) project. RP was funded by the UK Natural Environment Research Council (NERC) through the provision of funding for the National Centre for Earth Observation (NCEO, award reference no. NE/R016518/1) and the NERC-funded UKESM Earth sys-

tem modelling project (award no. reference NE/N017978/1). This work used JASMIN, the UK's collaborative data analysis environment (https://jasmin.ac.uk/, last access: 17/12/2024). The $H_2CO$ data products from OMI were generated at BIRA using level-1 data developed at NASA/KNMI. Level-2 and level-3 OMI $H_2CO$ developments are supported as part of the Sentinel-5 precursor TROPOMI level-2 project, funded by ESA and Belgian PRODEX (TRACE-S5P project). ES also acknowledges the use of ELM (University of Edinburgh, Edina, https://elm.edina.ac.uk, last access: 29/10/2024) for proofreading parts of the manuscript (abstract grammar and work count suggestions).



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
