# Peer review of "Biosphere-atmosphere related processes influence trace-gas and aerosol satellite-model biases."

_EGUsphere, 2024_

## Author Comment (AC1)

Dear editor and reviewers,

We thank the reviewers for their valuable feedback. We include the comments below (reviewer comments in bold), followed by our responses (regular black font) and text illustrating the changes in the manuscript (in italics, where red and blue represent deletions and additions, respectively). We have numbered the reviewer comments to help with the clarity of the response, e.g., in cases where both reviewers had similar feedback. Line numbers in the reviewer comments are from the submitted manuscript, while the line numbers in the responses refer to the tracked changes document.

Reviewer 1

1.  Major comments:

**You mention the indirect effects from land use change and BVOCs in the introduction and you investigate the changes in aerosol radiative effects between the simulations in Section 3.7. Why have you not calculated the changes in the indirect effects between the simulations? Even if you don't calculate them, I think it could be worth commenting on them in the paper and explaining why they are not calculated in this study.**

- Thank you for pointing out how this omission contrasts with the introduction of the paper. We calculated the cloud radiative effects but found the differences with all processes included to be mostly insignificant from a spatial perspective, although the global mean difference was significant. The results were subsequently omitted to keep the length of the paper down. We have now added text in the manuscript on the change in indirect effects when all processes but land cover are included, as well as a figure showing the differences in the supplement (Figure S10). Note the updated value in lines 598-610 (+0.163 W m$^{-2}$) is due to misreading the table with the source data from d'Andrea et al. (2015), an error that was spotted during the CRE updates.
    - In the methods:
        - Line 208: *2.4.2 Quantifying impacts on the aerosol direct  and indirect effects*
        - Lines 210-213: *and Cloud Radiative Effect (CRE; due to changes in cloud properties) between the simulations that include all processes with or without land cover (Exp_CS2mLC and Exp_CS2m, respectively) and the baseline simulation. We follow the recommendation of Ghan (2013) and calculate the DRE and CRE using*
        - Line 216: *CRE = ($F_{clean}$ − $F_{clear,clean}$)*
        - Line 218-219: *excluding scattering and absorption by aerosols, and $F_{clear,\ clean}$ is the net downward TOA radiative flux excluding scattering and absorption by aerosols and clouds.*

○ Additions to section 3.7:
- Line 564: *Lastly, we explore the aerosol-radiation and aerosol-cloud interactions in UKESM1.1*
- Line 567-569: *A similar process is followed to calculate the cloud radiative effect (CRE), which is defined as the difference in the net downward TOA radiative flux due to cloud properties and cloud cover (section 2.4.2). In this case, a negative DRE or CRE indicates greater outgoing radiation at the TOA.*
- Line 598-610: *The  present-day global average CRE in the standard model ($-24.54 \pm 0.02$ W m$^{-2}$) is greater than the DRE ($-1.73$ W m$^{-2}$). The impact of process representation is also greater in absolute terms: $-0.81 \pm 0.002$ W m$^{-2}$ without the observation-based land cover. Consistent with the changes in DRE, the impacts are greater in the SW. However, the influence of the biosphere-atmosphere processes is smaller in relative terms on the CRE than the DRE. The negative CRE is increased only by around 3 %, compared with the 8 to 10 % decrease in the magnitude of the DRE. Despite the significant change in the global mean value, the present-day CRE was not found to be significantly affected by the biosphere-atmosphere processes for most locations (Fig. S10).*

    *The changes in the present-day global aerosol DRE and CRE, due to the impacts of biosphere-atmosphere processes, calculated here ($+0.14$ W m$^{-2}$ to $+0.17$ W m$^{-2}$ for DRE and around $-0.81$ W m$^{-2}$ for CRE) are greater than or equivalent to the radiative forcing from aerosol-radiation interactions associated with pre-industrial to present-day land cover changes and their impacts on BVOC emissions as found in previous studies. For example, D'Andrea et al. (2015) find an aerosol direct effect of $+0.05$ to $+$$0.163$ W m$^{-2}$ and an aerosol indirect effect of $-0.008$ to $-0.056$ W m$^{-2}$ when comparing simulations with BVOC emissions for the years 1000 and 2000, (…)*

Minor comments:

2. **Page 4, line 112: How do the monoterpenes form Sec_OrgMt? Are there yields for specific oxidants or is it a set fraction of the monoterpene concentrations?**
   - We have modified the relevant sentences (line 111 to 118) to clarify this point: *"In both chemistry schemes,  the reaction of monoterpenes with $O_3$, OH or $NO_3$ forms a condensable vapour (Sec_Org$_{MT}$, where MT stands for monoterpenes) at a molar yield of 0.13 of the reaction fluxes. Sec_Org$_{MT}$ contributes to aerosol mass in all modes through irreversible condensation onto pre-existing particles. (…) As other BVOCs are also known to contribute to particle growth, the Sec_Org$_{MT}$ yield is scaled by a factor of two (to 0.26) to account for these missing sources of SOA (Mulcahy et al., 2020), unless otherwise stated."*
3. **Page 15, line 328: Should "lower" in the sentence be higher?**

- We have rewritten the sentence (line 332-333) for clarity:  *In DJF, substantial increases occur at 1000 m altitude* over east Asia  , *when $O_3$ mixing ratios increase by over 100%.*

4. **Page 19 Figure 8: The third column with the emissions ratio have a somewhat confusing color scale. The color scale is used for differences in other plots and it seems a little out of place in these figures. Please consider changing it.**

   - I have updated the colour scale for the emissions ratio in Figure 8 to purple-orange to avoid the confusion with the difference plots.

5. **Page 20, Figure 9. The last part of the figure caption should be removed. And please add subplot labels.**

   - The notes in the figure caption have been removed and subplot labels have been added.

6. **Page 21, line 420-422. Please consider dividing this sentence into two sentences.**

   - I have updated the sentence (line 428-430) as follows:
     *In summary, we find the inclusion of organically-mediated BLN tends to decrease AOD in regions proximate to monoterpenes emissions and $H_2SO_4$ sources. Further, the inclusion of Sec_Org$_I$ increases AOD values when and where $C_5H_8$ emissions are substantial, but decreases AOD over regions where monoterpenes are the main organic aerosol precursor.*

7. **Page 27, line 529. What does Supplement 9 refer to?**

   - I have added the following clarifying statement (line 543-544):
     "*The impact of applying the satellite retrieval averaging kernels on the model HCHO data is shown in more detail on Fig. S9.*"

8. **Page 28, Figure 14. I find this figure hard to interpret. Would it not be easier to just make a figure of a comparison of the updated model version compared to the satellite data?**

   - We acknowledge Figure 14 is complex, although it highlights the relative differences which may be difficult to see in a comparison of the absolute model-satellite biases. Based on this comment and comment 9 from Reviewer 2, we have moved the relative bias change magnitude figure to the supplement (Fig. S8) and replaced it with figures showing the model-satellite differences for the baseline simulation and the simulation inclusive of all processes (Exp_CS2mLC) as suggested. This change involves the following updates to the text in section 3.6 (lines 526-558):
     -  *Figures 14, 15 and 16 compare* the new model  *biases to those identified for the baseline model version. The significant decrease in TC $C_5H_8$ values in the tropics (discussed in section 3.2 and*

*3.5) results in  a 75 % decrease in $C_5H_8$ bias magnitude values in these regions in DJF and MAM ( Fig. 14e-f, with the relative changes shown in Fig.  S8). However, the change in TC values is so large that  the TC $C_5H_8$ values are now too low in some regions, e.g. in the Congo and eastern South America, while the magnitude of the  previously identified negative bias in east South America in SON increases. (...)*

- *Despite the substantial increase in HCHO driven by the more complex chemistry mechanism (sections 3.2 and 3.5) the change in HCHO model-satellite differences is small, which may be due to the sensitivity of the satellite retrieval to different altitudes. The impact of applying the satellite retrieval averaging kernels on the model HCHO data is shown in more detail on Fig.  S9. Regional biases in HCHO increase in southern Africa and northern Australia in DJF and MAM (Fig.  15e,f) in response to updated EFs and land cover (sections 3.4 and 3.5) , while the magnitude of the bias shows small decreases over eastern China. Due to the vertical sensitivity and substantial regional decreases, the magnitude of the global mean model-satellite difference in the TrC HCHO increases slightly ((<3%). (...)*

- *(...) None of the processes ameliorate the negative bias over North America in JJA (Fig.  16g). The altered process impacts are more notable elsewhere.  The bias in AOD decreases by up to  50% over the  Congo, likely in response to both land cover  and emission factor changes (section 3.4).*

Reviewer 2

9. **Lines 252-262: Concerning the two-sampling methodologies, I am concerned that the 6-hourly method underestimates the isoprene column by up to 1.5 x $10^{16}$ molec cm$^{-2}$ over isoprene hotspots in the tropics (Fig S2). In some ways, I am not surprised since the model may sometimes be sampled nearly 3 hours away from the 13:30 LT satellite overpass (which also happens to coincide with peak**

isoprene emission)! At any rate, an underestimation of 1-1.5 x $10^{16}$ molec $cm^{-2}$ by the 6-hourly method compared to the 13:30 LT method is substantial over isoprene hotspots. To help address this concern, I would like the authors to comment in Section 3.6 how their satellite and model comparison would change had their sampling methodology used the model output from 13:30 and not the 6-hourly method. I would also think this could impact their HCHO comparison as well given that HCHO is a major oxidation product of isoprene.

- Thank you for raising this issue and highlighting the relevance for section 3.6. We have added the following text to address the impact of sampling time and the daily cycle. As suggested in comments 8 and 12 we have replaced figure 14 with figures showing the new model biases next to the original model-satellite differences. We hope this allows for an easier comparison against the methodological differences in the supplement.

  - Lines 536-539:  *Considering the 6-hourly sampling approach is thought to underestimate the simulated TC $C_5H_8$ (section 3.1), the magnitude of these negative biases is likely overestimated. The annual global mean difference between the simulated and satellite-observed TC $C_5H_8$ decreases by 50% when the new processes are included, suggesting the improvements in the tropics have a substantial impact on the global average bias.*

  - Lines 550-552: *It is worth noting that the 6-hourly sampling may also affect this result, as HCHO may increase in the afternoon following the daily cycle of emissions, e.g. of biogenic precursors, although recent results have shown small amplitudes in the daily TC HCHO cycle over forest sites (Fu et al., 2025).*

10. Lines 304-307: These lines give the impression that BVOC emissions are reducing atmospheric oxidation capacity, but Lines 334-338 suggest it is due to the choice of chemical mechanism between CS2 and ST (particularly with $HO_x$ recycling in CS2 that would increase OH). I realize they're interrelated, but I would like the authors to clarify in the paper if they think the reduced atmospheric oxidation capacity is primarily due to increased BVOC emissions or the chemical mechanism.

- We have added further discussion on the relationships between increased BVOC emissions, choice of chemistry mechanism and atmospheric oxidation capacity in lines 339-345 and 499-501:

  - *The significant increase in OH with the inclusion of $HO_x$ recycling in the CS2 mechanism (Exp_CS2) suggests that BVOCs in the standard model set-up using the ST mechanism (baseline) may deplete OH too readily in regions of peak biogenic emissions.*  *While an increase in BVOC emissions in either chemistry scheme will affect the atmospheric oxidation capacity, the BVOC reactions are a greater oxidant sink in the ST scheme. This is due to the lack of $HO_x$ recycling in the $C_5H_8$ chemistry and, possibly more importantly, the unrealistic oxidant loss through the reactions with monoterpenes. While monoterpenes react with $O_3$, OH and $NO_3$, the only product of these reactions is $Sec\_Org_{MT}$ in the ST mechanism, resulting in the oxidants being removed with no possibility of even partial regeneration.*

- *The strong response to changing the chemistry mechanism compared to other processes highlights the dominant effect of the choice of mechanism on the atmospheric chemistry, including the atmospheric oxidation capacity.*

11. **Lines 438-439: Could you show a figure in the supplement showing where the loss of crops and pastures in favor of high emitting grasses is taking place? It just seems odd to me that an observational dataset would simplify down like that.**
   - A figure showing the changes in C3 and C4 grass PFTs and, separately, the crop and pasture PFTs has been added with the following text to the supplement:
     - *Figure S6 shows the differences in the fraction of grasses, as well as crops and pastures, between the ESA CCI land cover data and the model-derived plant functional type (PFT) distribution. The original land cover types in the observational dataset are not identical to the model PFTs, resulting in some simplifications to enable the data to be used as model input.*
   - The new figure in the supplement is referred to in line 447 of the manuscript:
     - *Therefore, the use of the observation-based land cover not only affects the spatial distribution of the PFTs, but also results in the complete loss of crops and pastures in favour of high emitting grasses (Fig. S6).*

12. **Figure 14: This is one of the main summary plots of the paper, but I found it confusing to report data as "new bias as a percentage of old bias." Why not compare Exp_CS2mLC to the satellite data directly for the different seasons since then the reader can see how far off in sign and magnitude the model (Exp_CS2mLC) is compared to satellite? The original Figure 14 could then be moved to supplemental for readers interested in comparison back to the baseline model.**
   - Thank you for highlighting the challenges associated with Figure 14. The original Fig. 14 has been moved to the supplement and replaced with direct comparisons. The associated updates to the text in section 3.6 are included in detail in the response to comment 8.

13. **Line 522-523: Is the magnitude of the global mean difference for a particular season or annual?**
   - This is the change in the magnitude of the annual global mean difference. The text has been expanded accordingly in line 537:
     - *The annual global mean difference between the simulated and satellite-observed TC C5H8 decreases by 50%, suggesting the improvements in the tropics have a substantial impact on the global average bias.*

14. **Line 601: In addition to missing sources of HCHO, do the authors think that the standard setup just does not get oxidants correct (considering that the standard mechanism does not have HOₓ recycling)? That could also explain why the standard model greatly underestimates HCHO.**
   - Yes, we believe there are issues with the representation of oxidants in the ST chemistry mechanism. We acknowledge that the CS2 scheme is also not perfect. To highlight the challenges associated with the ST scheme we have added the following sentence (lines 633-634):

> ▪ *Further, in the ST mechanism, $C_5H_8$ chemistry is substantially simplified, while the monoterpenes reactions, of which the only product is Sec_Org$_{MT}$, are a complete oxidant sink, leading to inaccuracies in the representation of oxidants with implications for HCHO formation.*

**Technical Corrections:**

15. **Line 245: "than" instead of "then"**
    - Thank you for your careful and detailed review of our manuscript. This error has been corrected.

16. **Line 365: "emissions" instead of "emisisons"**
    - This error has been corrected.

17. **Fix Figure 9 caption**
    - This error has been corrected, as described in the response to comment 5.

18. **Fix Line 419 formatting**
    - This issue has been fixed.

19. **Line 454: Double-check figure subsets. Figures 11a and 11b do not use the new EFs, correct?**
    - This is correct. Figure 11b should have been 11d in the text. This has now been updated in line 462:
        - *(compare Fig. 11a,b and 11d,e).*

20. **Line 501: "section" instead of "seciton"**
    - This error has been corrected.

21. **Line 505: Awkward phrasing of "...how processes involved in the biosphere impacts on atmospheric chemistry and aerosols have a wide range of effects..."**
    - The text has been updated as follows (lines 520-523):
        - In this section we outline  the impact of the biosphere-atmosphere processes  on these biases. *The simulation inclusive of all processes (Exp_CS2mLC)  is compared to the satellite data,  so that these differences can also be compared to the original biases found for UKESM1.1 (section 3.1).*

22. **Supplemental: Confusing to have section headers different than the figure numbers. For example, Figure S6 appears in S8.**
    - We have removed the numbering of the supplement sections and refer directly to the supplemental table or figure in the manuscript to avoid the confusion.